# Automatic Image-Level Morphological Trait Annotation for Organismal Images

**Vardaan Pahuja**[1]    **Samuel Stevens**[1]    **Alyson East**[2]    **Sydne Record**[2]    **Yu Su**[1]
[1]The Ohio State University        [2]University of Maine
`pahuja.9@osu.edu`

## Abstract

Morphological traits are physical characteristics of biological organisms that provide vital clues on how organisms interact with their environment. Yet extracting these traits remains a slow, expert-driven process, limiting their use in large-scale ecological studies. A major bottleneck is the absence of high-quality datasets linking biological images to trait-level annotations. In this work, we demonstrate that sparse autoencoders trained on foundation-model features yield monosemantic, spatially grounded neurons that consistently activate on meaningful morphological parts. Leveraging this property, we introduce a trait annotation pipeline that localizes salient regions and uses vision-language prompting to generate interpretable trait descriptions. Using this approach, we construct BIOSCAN-TRAITS, a dataset of 80K trait annotations spanning 19K insect images from BIOSCAN-5M. Human evaluation confirms the biological plausibility of the generated morphological descriptions. We assess design sensitivity through a comprehensive ablation study, systematically varying key design choices and measuring their impact on the quality of the resulting trait descriptions. By annotating traits with a modular pipeline rather than prohibitively expensive manual efforts, we offer a scalable way to inject biologically meaningful supervision into foundation models, enable large-scale morphological analyses, and bridge the gap between ecological relevance and machine-learning practicality.[1]

## 1 Introduction

The accelerating biodiversity crisis demands rapid advancement in our understanding of ecosystem function and species' responses to environmental change. While taxonomic identification answers the question "what species is this?", it fails to explain *why* organisms succeed or fail under changing conditions. Morphological traits (the measurable physical characteristics of organisms) provide this critical mechanistic link, predicting with remarkable accuracy how species interact with their environment (Díaz et al., 2016; Kennedy et al., 2020; McGill et al., 2006). Morphological traits can predict species' ecological niches and functions with up to 85% accuracy (Pigot et al., 2020), offering insights into resource utilization and potential responses to disturbance. Despite their paramount importance, trait data remains trapped in an analog bottleneck: millions of biological specimens and images exist in collections worldwide, but extracting standardized trait measurements requires painstaking manual work by domain experts (Violle et al., 2007), rendering large-scale trait-based ecology virtually impossible.

Measuring even simple characters such as body length or tibia ratio still takes minutes per specimen despite modern digitization techniques (Hardisty et al., 2022). Natural-history institutions curate 3B+ specimens, so a full trait census would consume person-centuries of expert labour (Nelson & Ellis, 2019). Protocols differ by taxon (wing chord for birds, elytral lengths for beetles, sepal length for plants, etc) and this heterogeneity, combined with observer subjectivity, introduces systematic bias that complicates data synthesis (Heberling, 2022). Even when traits are quantified, they often remain in notebooks or image captions, invisible to machine pipelines, leaving a global "trait data desert" that blocks large-scale trait ecology studies.

---

[1]Code and data are available at `osu-nlp-group.github.io/sae-trait-annotation/`.

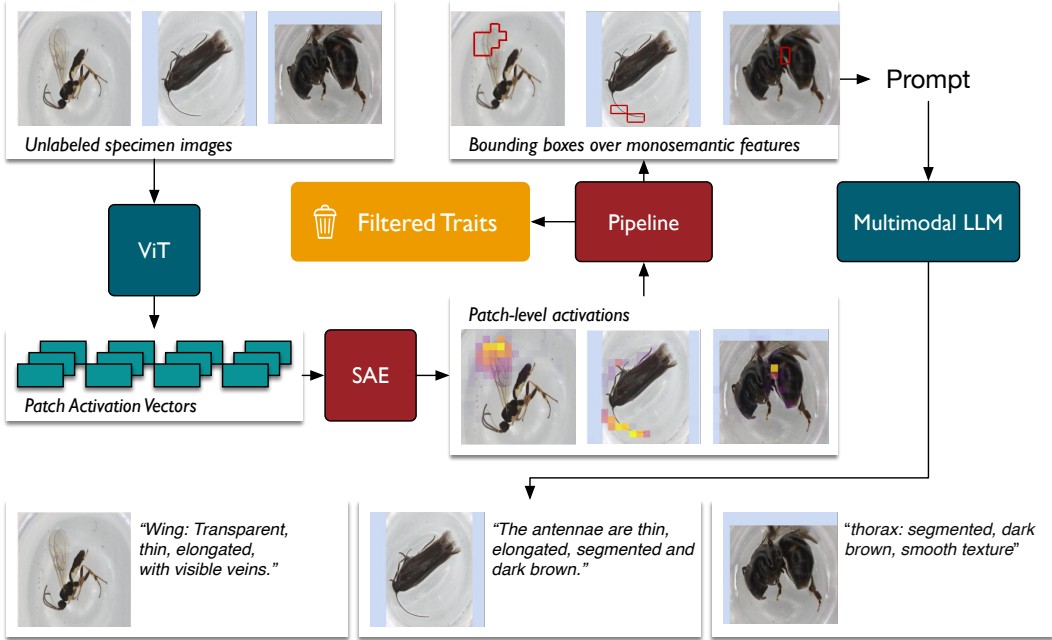

Figure 1: Given an input specimen image, we first compute dense visual representations using an off-the-shelf backbone (*e.g.*, DINOv2). These features are passed through a pre-trained sparse autoencoder (SAE), which identifies high-activation latent units corresponding to semantically meaningful regions (Algorithm 1). We extract the spatial masks associated with these activations and overlay them on the original image to localize trait-relevant boxes. Finally, a multimodal language model (MLLM) is prompted with the annotated image to generate fine-grained morphological trait descriptions. **This results in a large-scale, automatically labeled image-level trait dataset.**

Automating trait mining pushes ML into a worst-case regime. First, biology's cross-taxon heterogeneity means the feature manifold warps whenever one moves from, say, angiosperm leaves to wasp antennae; He et al. (2024) lists this taxonomic domain shift as the single largest unsolved barrier to reliable pipelines. Digitized specimens further exhibit uncontrolled pose, preservation artifacts, and background clutter, factors that amplify distribution shift and explode the sample complexity demanded by supervised learning. Second, a systematic review of 50+ herbarium-vision papers finds that apparently "simple" tasks (leaf area, margin type) still need bespoke augmentation recipes and hyper-parameter sweeps for every dataset, with no method transferring cleanly across collections (Hussein et al., 2022). Third, even mature semi-automated tools, such as *Inselect* (Hudson et al., 2015) for drawer segmentation, end up handing users a GUI for redrawing boxes; human operators spent 108 seconds per image correcting model outputs. Together, these observations show that standard supervised learning struggles when labels are scarce, morphology is non-stationary, and objects occupy only tiny, variable parts of the frame—precisely the conditions that trait ecology presents.

Our key insight is recognizing that **sparse autoencoders (SAEs) can be used as interpretable part-detectors for trait extraction**. A sparse autoencoder learns, from unlabeled data, a dictionary of latent units that can linearly reconstruct frozen foundation-model embeddings while enforcing two pressures: (i) *sparsity*: only a few units fire for any image, and (ii) *non-negativity*: activations cannot cancel each other. These constraints push each latent unit toward a single, reusable visual cause rather than a mixture of unrelated cues. In practice, training an SAE over pre-trained image features produces units whose activations map back onto tight, spatially coherent regions such as "hind-leg femur band," "dorsal eye stripe," or "apical leaf tip." (see §4.4 for example visualizations) After training, we can (1) isolate just the pixels that define a candidate trait, (2) visually indicate the relevant area, and (3) describe those areas with a vision-language model. To focus on truly diagnostic parts, we introduce a species-contrastive ranking: a unit is valuable when it fires strongly for a target species but remains almost silent for closely related species. High-ranked units, therefore, highlight

precisely the salient, fine-scale structures that taxonomists record as traits, making the SAE an ideal front end for our trait-distillation pipeline.

We instantiate these ideas in a three-step, concrete, trait-labeling pipeline (Figure 1) and apply it to the BIOSCAN-5M insect corpus (Gharaee et al., 2024):

1. We rank SAE units by a species-contrastive score that privileges activations that are strong for a focal species yet weak for its congeners.
2. High-score masks are boxed into tight patches.
3. Each patch is prompted to a large multimodal large language model (Qwen2.5-VL-72B) with a lightweight template.

Because the SAE provides locality and taxonomic focus before any language model is consulted, the MLLM's task is far easier: "describe this part" rather than "describe the whole scene", which sharply reduces hallucinations and background leakage. While BIOSCAN-5M provides the large-scale, species-labeled data for our experiments, the pipeline itself only requires images with taxonomic labels. Such supervision is widely available in many other domains (*e.g.*, iNaturalist (Horn et al., 2018), TreeOfLife (Stevens et al., 2024), Caltech-UCSD Birds-200-2011 (Wah et al., 2011)). These resources span plants, birds, fungi, and other taxa, making our approach broadly applicable for converting labeled image repositories into rich, interpretable trait annotations. Using the best-performing configuration, we label 19K images with 80K morphological trait descriptions (averaging 4.2 traits per image), yielding the BIOSCAN-TRAITS dataset. To evaluate robustness and design sensitivity, we conduct a comprehensive ablation study, systematically examining how individual design choices influence the quality of the resulting trait descriptions. As an initial validation, we fine-tune BioCLIP (Stevens et al., 2024; Gu et al., 2025), a biologically grounded vision–language foundation model on BIOSCAN-TRAITS and observe improved zero-shot species classification on an in-the-wild benchmark, highlighting the downstream potential of trait-level supervision.

In summary, we contribute (i) a species-contrastive SAE-and-MLLM-based algorithm that turns unsupervised images into high-fidelity, spatially grounded trait labels and (ii) BIOSCAN-TRAITS, a large, open, image-trait dataset, and (iii) an initial downstream evaluation showing that fine-tuning a foundation model on BIOSCAN-TRAITS improves zero-shot species classification on an in-the-wild benchmark. By using a modular pipeline for trait annotation instead of expensive manual labeling, we provide a scalable way to incorporate biologically meaningful supervision into foundation models, support large-scale morphological analyses, and narrow the gap between ecological relevance and practical machine-learning workflows.

## 2 RELATED WORK

**Sparse Autoencoders.** Sparse autoencoders (SAEs) (Makhzani & Frey, 2014; 2015) have proven effective for uncovering disentangled and human-interpretable latent factors in high-dimensional representations (Stevens et al., 2025). Prior work has shown the utility of SAEs to learn improved image (Makhzani & Frey, 2014; 2015) and word representations (Subramanian et al., 2018). To enhance feature disentanglement and interpretability, several architectural variants have been proposed, including top-$k$ activation mechanisms (Bussmann et al., 2024) and multi-layer Matryoshka encoders designed to promote hierarchical concept structure (Bussmann et al., 2025). SAEs have also been applied to the internal activations of transformer-based language models, where they reveal latent units aligned with semantically meaningful and interpretable concepts (Yun et al., 2021; Bricken et al., 2023; Gao et al., 2025; Templeton et al., 2024). Recent work demonstrates that, when trained on embeddings from large pretrained models, SAEs can produce monosemantic features, latent units that respond consistently to a single semantic concept (Templeton et al., 2024; Pach et al., 2025). In this work, we extend these insights to the domain of biological vision, using SAEs to construct a dataset of fine-grained morphological traits from organismal images.

**Fine-grained Visual Recognition.** Fine-grained visual recognition (FGVR) (Lin et al., 2015) aims to distinguish subordinate categories with small inter-class variation but large intra-class variation, where the most discriminative cues are often subtle and localized (*e.g.*, texture, shape, or color patterns). As a result, FGVR models are particularly vulnerable to background correlations, viewpoint and pose changes, and the scarcity of expert annotations (Beery et al., 2018). A major line of work therefore seeks to localize discriminative regions without dense part supervision, using either weak

supervision (Hu et al., 2019) or self-supervised consistency signals (Huang et al., 2020; Wu et al., 2022). In real-world settings, FGVR must further contend with distribution shifts across environments, motivating benchmarks and methods that emphasize out-of-distribution (OOD) generalization (Beery et al., 2020; Koh et al., 2021; Pahuja et al., 2024). More recently, language has emerged as a useful interface for fine-grained semantics: models extract or generate part-level attributes and leverage MLLM reasoning to better align visual evidence with fine-grained category names (Liu et al., 2024). In this context, our work uses sparse autoencoders to automatically extract morphological traits, providing trait-level supervision that improves fine-grained visual recognition.

**Morphological Trait Extraction.** Traditionally, morphological analysis has relied on manual measurements and qualitative trait descriptions—a process that is labor-intensive, time-consuming, and dependent on domain expertise (Hunt & Pedersen, 2025). While these methods offer valuable insights, they are inherently difficult to scale to large datasets. Recent approaches have begun to automate trait extraction by leveraging representation learning. For instance, Hoyal Cuthill et al. (2019) used a convolutional triplet network to map images into a phenotypic embedding space, enabling quantitative similarity measures and phenotypic tree reconstruction from purely visual data. More recent work has pushed further: deep models that segment relevant image regions (*e.g.*, in herbarium scans (Ariouat et al., 2025)) or learn latent representations (*e.g.*, via VAEs (Tsutsumi et al., 2023)) show that rich morphological information can be captured without hand-engineered features. A key challenge, however, is developing models that remain robust to digitization artifacts and background clutter, while also offering interpretability so that ecologists can identify which morphological features drive predictions. Our work leverages SAEs to automatically extract morphological traits in BIOSCAN (Gharaee et al., 2024) specimen images. We posit that such trait-level supervision can enhance the robustness and generalizability of MLLMs for fine-grained taxonomic classification.

## 3 METHODOLOGY

### 3.1 BACKGROUND

Sparse autoencoders (SAEs) transform dense representations into sparse encodings, where each unit ideally corresponds to an interpretable latent factor. Given a dense input vector $\boldsymbol{z} \in \mathbb{R}^d$ from an intermediate layer of a vision transformer, the autoencoder maps $\boldsymbol{z}$ to a high-dimensional sparse representation $g(\boldsymbol{z})$, from which $\boldsymbol{z}$ is subsequently reconstructed. This decomposition reveals structured latent factors while preserving the original information content. We use ReLU autoencoders (Bricken et al., 2023; Templeton et al., 2024) for our experiments.

$$\boldsymbol{u} = \boldsymbol{W}_e(\boldsymbol{z} - \boldsymbol{b}_d) + \boldsymbol{b}_e, \tag{1}$$

$$g(\boldsymbol{z}) = \text{ReLU}(\boldsymbol{u}), \tag{2}$$

$$\tilde{\boldsymbol{z}} = \boldsymbol{W}_d \, g(\boldsymbol{z}) + \boldsymbol{b}_d, \tag{3}$$

where $\boldsymbol{W}_e \in \mathbb{R}^{n \times d}$, $\boldsymbol{b}_e \in \mathbb{R}^n$, $\boldsymbol{W}_d \in \mathbb{R}^{d \times n}$, and $\boldsymbol{b}_d \in \mathbb{R}^d$. Here, $\boldsymbol{W}_e \in \mathbb{R}^{n \times d}$ denotes the SAE encoder matrix that maps the dense backbone representation $\boldsymbol{z} \in \mathbb{R}^d$ to the pre-activation latent vector $\boldsymbol{u} \in \mathbb{R}^n$, and $\boldsymbol{W}_d \in \mathbb{R}^{d \times n}$ denotes the decoder matrix that maps the sparse code back to the reconstructed representation $\tilde{\boldsymbol{z}} \in \mathbb{R}^d$. The encoder and decoder also include bias terms: $\boldsymbol{b}_e \in \mathbb{R}^n$ and $\boldsymbol{b}_d \in \mathbb{R}^d$, respectively.

The training objective minimizes the reconstruction error while encouraging sparsity in the latent representation:

$$\mathcal{J}(\phi) = \|\boldsymbol{z} - \tilde{\boldsymbol{z}}\|_2^2 + \alpha \, \mathcal{R}(g(\boldsymbol{z})), \tag{4}$$

where $\mathcal{R}$ denotes the sparsity regularizer and the sparsity coefficient ($\alpha$) controls the trade-off between sparsity and reconstruction. We use DINOv2-base (Oquab et al., 2024) as the feature backbone to extract dense visual representations from specimen images (see ablations in Appendix E).

---

**Algorithm 1:** Salient Trait Extraction from Sparse Autoencoder Activations

---

**Input:** Species-labeled dataset $\mathcal{D} = \{(x_i, y_i)\}_{i=1}^{N}$
    Trained sparse autoencoder $f_\theta$
    Activation threshold $t_{\text{activation}}$
    Normalized frequency threshold $t_{\text{freq}}$
**Output:** Set of salient traits $\mathcal{T}_{\text{distinct}}$ for each species

1   Initialize counters $C_{\text{species}}$ and $C_{\text{genus}}$ as empty maps;
2   **foreach** $(x_i, y_i) \in \mathcal{D}$ **do**
3     $z_i \leftarrow f_\theta(x_i)$ ;                          `// Sparse latent vector`
4     $\mathcal{Z}_i \leftarrow \{z_j \mid z_i[j] > t_{\text{activation}}\}$;
5   **foreach** *trait z* **do**
6

$$C_{\text{species}}[s][z] = \sum_{i:y_i=s} \mathbf{1}[z \in \mathcal{Z}_i] \qquad C_{\text{genus}}[g][z] = \sum_{i:\text{genus}(y_i)=g} \mathbf{1}[z \in \mathcal{Z}_i]$$

7   **foreach** *species s and its genus g* **do**
8     **foreach** *trait z* **do**
9       $f_s(z) \leftarrow \frac{C_{\text{species}}[s][z]}{\sum_{z'} C_{\text{species}}[s][z']}$;
10      $f_g(z) \leftarrow \frac{C_{\text{genus}}[g][z]}{\sum_{z'} C_{\text{genus}}[g][z']}$;
11   Initialize $\mathcal{T}_{\text{distinct}} \leftarrow \{\}$;
12   **foreach** *species s with genus g* **do**
13     $\mathcal{T}_s \leftarrow \{z \mid f_s(z) > t_{\text{freq}} \wedge f_g(z) > t_{\text{freq}} \wedge f_s(z) > f_g(z)\}$;
14     $\mathcal{T}_{\text{distinct}}[s] \leftarrow \mathcal{T}_s$;
15   **return** $\mathcal{T}_{distinct}$

---

## 3.2   DATASET GENERATION

We use the high-activation latents (with values above a certain threshold $t_{activation}$) to generate descriptions of salient morphological traits in species images. The trait extraction procedure consists of the following steps:

1. **Sparse Activation Computation**: For each image in the BIOSCAN-5M dataset annotated at the species level, we compute its sparse latent representation using the trained autoencoder.
2. **Trait Selection via Activation Thresholding**: From the full set of activated latent features for a given sample, we retain only those whose activation values exceed a predefined threshold (denoted by $t_{\text{activation}}$), indicating salient trait expression.
3. **Taxonomic Trait Aggregation**: We then compute the frequency distribution of activated traits at both the species and genus levels across the dataset.
4. **Trait Filtering by Prevalence**: Within each taxonomic rank, we retain only those traits whose normalized frequency, computed as the ratio of trait occurrences to the total number of trait activations for the taxon, exceeds a predefined threshold (denoted by $t_{\text{freq}}$). This filtering step mitigates noise and retains consistently expressed traits.
5. **Salient Trait Identification**: We identify salient morphological traits for a species as the ones expressed in a significantly higher proportion within that species than across its corresponding genus, indicating taxon-specific salience.

Algorithm 1 illustrates the procedure for selection of salient traits in detail. Given these traits, we prompt multimodal language models to query the morphological trait descriptions (Figure 1). Prompt templates are provided in Appendix C, and dataset statistics are in Table D.5. Additional dataset examples are shown in Appendix G. We also discuss ecology applications in Appendix I.

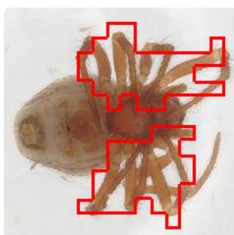 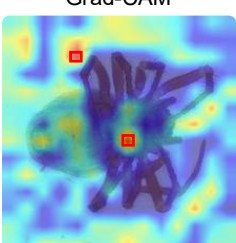

BIOSCAN-TRAITS     Grad-CAM

**Trait Descriptions**
**BIOSCAN-TRAITS**: Leg: Thin, elongated, light brown, segmented.
**Grad-CAM**: Head: Highlights the cephalothorax, emphasizing its rounded shape and the presence of sensory organs. Legs: Focuses on the anterior legs, showcasing their robust structure and segmentation. - Body: Emphasizes the abdomen, noting its segmented appearance and the presence of spinnerets at the posterior end.

Figure 2: Comparison of trait localization for *Thymoites guanicae*. **BIOSCAN-TRAITS (left)** generates interpretable trait descriptions that are tied to clear, specific anatomical structures. In contrast, **Grad-CAM (center)** produces diffuse heatmaps that highlight broad body areas without species-level disentanglement.

## 4 EXPERIMENTS

### 4.1 SPARSE AUTOENCODER TRAINING

We use the BIOSCAN-5M dataset (Gharaee et al., 2024) for our experiments. BIOSCAN-5M is a comprehensive dataset of insect specimens with multiple modalities, including images, DNA barcodes, taxonomic, geographic, and size information. It contains insect images annotated at different levels of the taxonomic hierarchy, with $9.2\%$ of the samples annotated at species-level. While our experiments use BIOSCAN-5M as the large-scale, species-labeled dataset, the method itself only needs image collections paired with taxonomic labels—supervision that is common across many repositories (e.g., iNaturalist (Horn et al., 2018) and TreeOfLife (Stevens et al., 2024)). Such datasets cover plants, birds, fungi, and many other groups; therefore, the pipeline is broadly applicable and can scale to transform species- or genus-labeled biological image archives into rich, interpretable trait-level annotations.

We train the sparse autoencoder on the entire set of images in BIOSCAN-5M, while the trait generation pipeline uses the subset with species-level labels. The complete hyperparameter setup is given in Table D.4 in the Appendix D.

### 4.2 COMPARISON WITH GRAD-CAM

We compare our pipeline to using traditional feature visualization approaches like Grad-CAM (Selvaraju et al., 2017) for obtaining saliency maps and then forwarding to the MLLM for trait generation (Figure 2). While Grad-CAM can highlight salient regions for a given class label, it lacks trait-level disentanglement, *i.e.*, its heatmaps typically blend multiple anatomical cues, making it difficult for an MLLM to generate precise, interpretable trait descriptions. Moreover, Grad-CAM activations are not species-discriminative, often capturing features shared across related taxa (genus or family level), whereas our SAE-based approach explicitly isolates species-specific, monosemantic neurons tied to fine-grained traits.

### 4.3 DATASET ABLATIONS

We conduct a series of ablation studies to evaluate the impact of key design choices on the accuracy and plausibility of trait annotations. For each configuration, we randomly sample 30 trait descriptions and evaluate them using a five-point rubric. Three domain experts independently rated the samples. We apply per-rater mean normalization to ratings, rescaling each annotator's scores so that their personal mean equals the global mean (Riley et al., 2024; Kirk et al., 2024). This ensures that differences in individual scale usage (*e.g.*, consistently harsh or lenient raters) do not skew aggregated results. The evaluation rubric is given in Appendix F.

**Comparison with MLLM-only baseline.** As a baseline, we prompt a multimodal large language model (MLLM) with just the specimen image(s) without the trait localization and request a description of salient morphological traits (Figure 3). We compare this to our SAE-guided trait extraction pipeline,

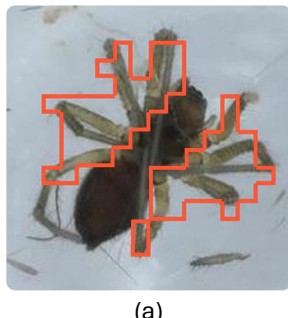 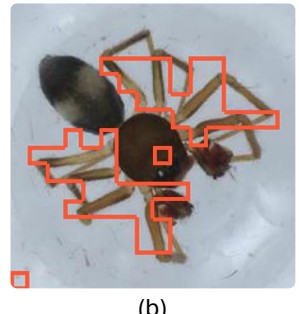 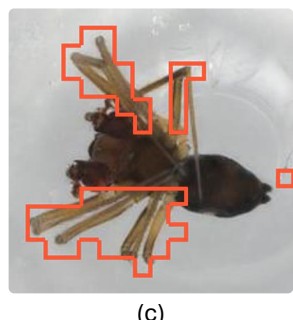

(a)                   (b)                 (c)

> **MLLM**: Common Morphological Traits Across All Three Images: 1. Head:- *Shape*: Rounded. - *Color*: Dark brown or black. - *Texture*: Appears smooth with no distinct markings. 2. Thorax: - *Shape*: Oval and slightly elongated. - *Color*: Lighter than the head, ranging from light brown to beige. - *Texture*: Smooth, with no visible segmentation. 3. Abdomen: - *Shape*: Rounded and bulbous. - *Color*: Dark brown or black, similar to the head. - *Texture*: Smooth, with no distinct markings. 4. Legs: - *Number*: Eight legs are visible in all images. - *Shape*: Thin and segmented. - *Color*: Light brown, matching the thorax.
> **MLLM + SAE**: **Leg**: Thin, elongated, light brown, segmented.

Figure 3: Comparison of salient morphological trait description generation using a just MLLM vs. MLLM + SAE ($t_{\text{freq}} = 1e-2$) for *Agyneta straminicola*. Each red box highlights a region selected by SAE neurons with high activation, indicating regions used for prompting the MLLM + SAE. The use of SAE helps MLLMs focus on salient morphological traits rather than general descriptions of all body parts.

Table 1: Incorporating latent-specific patches significantly improves the quality of trait descriptions. Including multiple images in the prompt encourages MLLMs to focus on the traits common across all images, at the cost of more tokens per query. Using multiple images with SAE-extracted bounding boxes leads to improved precision, as better ratings indicate. We report both raw and mean-normalized ratings. The experimental setup uses Qwen2.5-VL-72B as MLLM, a normalized frequency threshold of ($t_{\text{freq}}$) = 3e−3, and 1,000 input images.

| Method | # Images | # Tokens | # Images | # Traits | Avg. Raw Rating | Avg. Rating |
|---|---|---|---|---|---|---|
| MLLM | 1 | 413 | – | – | 3.01 | 3.00 |
| MLLM | 3 | 940 | – | – | 3.12 | 3.15 |
| MLLM + SAE | 1 | 411 | 460 | 9,435 | 3.92 | 3.84 |
| MLLM + SAE | 3 | 1,072 | 460 | 7,897 | **4.01** | **3.91** |

which localizes trait-relevant regions via sparse latent activations (Table 1). Incorporating latent-specific patches leads to a substantial improvement in description quality: the average human rating increases from 3.15 to 3.91 in the multi-image setting, highlighting the benefits of spatial grounding provided by the sparse autoencoder for fine-grained trait extraction.

**Multiple vs. Single Image per Latent.** We investigate the effect of varying the number of input images on trait quality by comparing single-image against 3-image prompts to the multimodal language model (Table 1). Providing multiple images of the same species encourages the model to focus on consistent, shared morphological features while suppressing spurious or image-specific traits. This consensus-driven trait extraction leads to improved precision, as reflected by an increase in the average human rating from 3.84 to 3.91, albeit at the cost of higher token usage per query. A similar trend holds for the MLLM-only baseline.

Additionally, we do a qualitative analysis of the morphological trait descriptions generated by both approaches (Figure 4). Using a single image often leads to trait descriptions that overfit to idiosyncratic visual details of that instance, frequently summarizing multiple anatomical regions, as seen in the example where both the legs and abdomen are described together. This broad coverage can dilute trait precision and obscure what is taxonomically distinctive. In contrast, prompting the model both with multiple images and latent-specific regions encourages it to extract traits that are

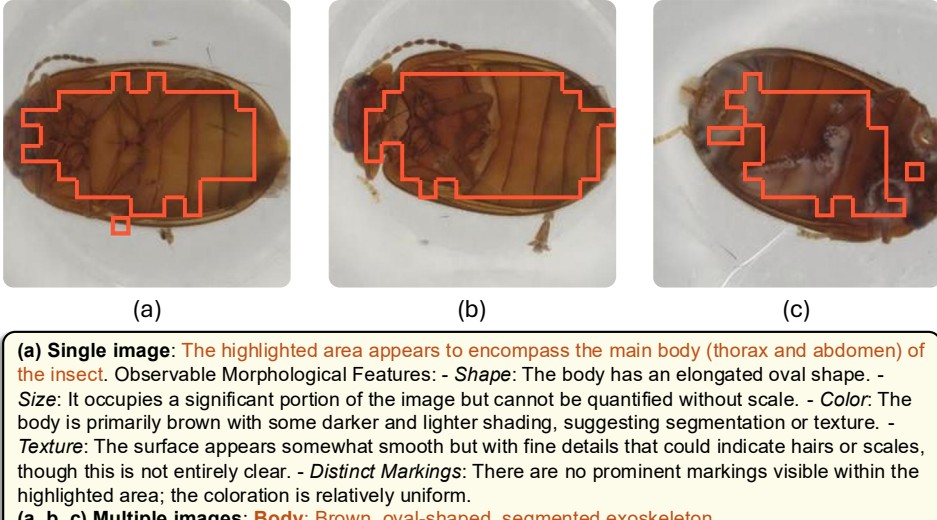

(a) **Single image**: The highlighted area appears to encompass the main body (thorax and abdomen) of the insect. Observable Morphological Features: - *Shape*: The body has an elongated oval shape. - *Size*: It occupies a significant portion of the image but cannot be quantified without scale. - *Color*: The body is primarily brown with some darker and lighter shading, suggesting segmentation or texture. - *Texture*: The surface appears somewhat smooth but with fine details that could indicate hairs or scales, though this is not entirely clear. - *Distinct Markings*: There are no prominent markings visible within the highlighted area; the coloration is relatively uniform.
(a, b, c) **Multiple images**: **Body**: Brown, oval-shaped, segmented exoskeleton.

Figure 4: Comparison of salient morphological trait description generation using a single image vs. three images for *Contacyphon ochraceus*. Each red box highlights a region selected by SAE neurons with high activation, indicating regions used for prompting the MLLM + SAE. The use of multiple images yields a concise and taxonomically meaningful output, isolating traits with clearer morphological grounding.

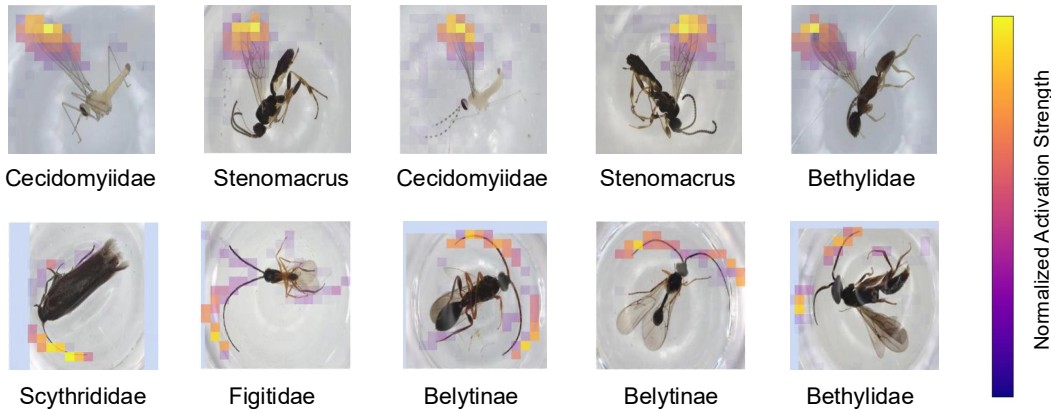

Figure 5: Neurons 4852 and 13860 in SAE get activated at the wings and antennae of insects, respectively. The labels denote the highest annotated taxonomic level. Additional examples are shown in Appendix J.

visually consistent across specimens. This consensus constraint filters out incidental details and leads to more focused, high-precision descriptions (*e.g.*, isolating just the leg features). As shown in Figure 4, the multi-image setup yields a concise and taxonomically meaningful output, isolating traits with clearer morphological grounding and higher inter-image agreement.

**SAE Quality.** We investigate the sparse autoencoder's inherent tradeoff between reconstruction error and sparsity and its downstream impact on morphological trait generation (Table 2). Specifically, we compare performance across varying values of the sparsity regularization coefficient ($\alpha$), which controls the $L_0$-sparsity of the latent representation. We observe that lower sparsity (*i.e.*, smaller $\alpha$, larger $L_0$) consistently yields better performance across both values of the normalized frequency threshold $t_{\text{freq}}$ (Figure 6). We hypothesize that reduced sparsity activates a broader set of latents per image, providing richer and more stable part proposals that better cover insect anatomy and reduce missed discriminative regions, which in turn improves the quality of the resulting trait descriptions.

Table 2: SAEs often trade off between reconstruction error (MSE) and sparsity ($L_0$). We investigate the effect of choosing between different balances of these errors. We find that lower sparsity performs better for both values of frequency threshold ($t_{\text{freq}}$). A lower value of the sparsity coefficient ($\alpha$) leads to lower MSE and thus better reconstruction. It improves the coverage of latents, leading to better recall. The experimental setup uses an input dataset of 1,000 images.

| Method | $\alpha$ | $t_{\text{freq}}$ | SAE MSE | SAE $L_0$ | # Images | # Traits | Avg. Rating |
|---|---|---|---|---|---|---|---|
| MLLM+SAE | 2e−4 | 1e−2 | 8.8e−3 | 1,081.1 | 60 | 60 | 3.84 |
| MLLM+SAE | 4e−4 | 3e−3 | 2.7e−2 | 690.4 | 460 | 7,897 | **3.91** |
| MLLM+SAE | 4e−4 | 1e−2 | 2.7e−2 | 690.4 | 20 | 20 | 3.58 |
| MLLM+SAE | 8e−4 | 3e−3 | 5.4e−2 | 242.2 | 458 | 3,060 | 3.87 |

Table 3: Effect of normalized frequency threshold ($t_{\text{freq}}$) on trait selection. We analyze how varying $t_{\text{freq}}$, which controls the minimum intra-species normalized frequency required to retain a latent feature, impacts trait extraction. Lower thresholds include all activated traits, while higher thresholds restrict output to only the most consistently expressed traits. Increasing $t_{\text{freq}}$ improves precision but reduces the number of extracted traits, reflecting a trade-off between coverage and specificity.

| Method | $t_{\text{freq}}$ | # Images | # Traits |
|---|---|---|---|
| MLLM+SAE | 3e−3 | 460 | 7,897 |
| MLLM+SAE | 6e−3 | 322 | 785 |
| MLLM+SAE | 1e−2 | 20 | 20 |

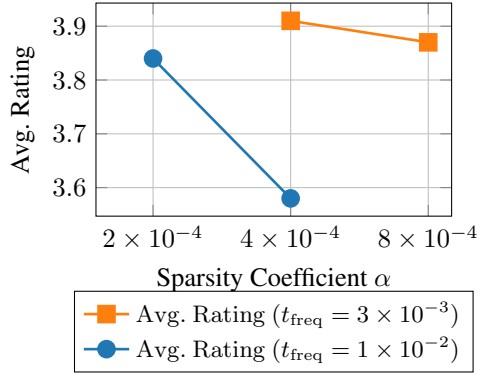

Figure 6: Variation of rating with different levels of SAE sparsity. A lower level of sparsity performs better for both values of frequency threshold $t_{\text{freq}}$.

**SAE Filtering.** We analyze the effect of the normalized frequency threshold $t_{\text{freq}}$ on the trait throughput using 1,000 input images and sparsity coefficient ($\alpha$) = $4e-4$. We observe that increasing $t_{\text{freq}}$ leads to a progressive reduction in the number of retained latent features (Table 3). This results in the selection of only the more consistently activated latents across a taxon, effectively narrowing the subset of input images that contribute to trait descriptions. Thus, $t_{\text{freq}}$ acts as a precision–recall knob: lower values yield broader trait coverage but more noise, while higher values emphasize dominant, taxonomically stable traits.

**MLLM Quality.** We compare Qwen2.5-VL-7B and Qwen2.5-VL-72B (Wang et al., 2024) for trait generation from latent-indexed patches. The larger 72B model yields higher human evaluation scores and better spatial grounding, avoiding false positive traits; see Appendix B for details.

## 4.4 NEURON ACTIVATION ANALYSIS

We analyze the top-activating neurons (or latent dimensions) in the SAE to investigate whether they correspond to meaningful morphological traits. Representative examples are shown in Figure 5. Notably, neuron 4852 consistently activates on insect wings, while neuron 13860 responds to antennae, suggesting that specific neurons in the sparse representation are aligned with semantically coherent, interpretable, and biologically plausible traits.

## 4.5 COST-OF-USE ANALYSIS

We next quantify the efficiency and cost of our pipeline and examine how well the SAE-guided prompting strategy transfers across different MLLMs. Table 4 reports runtime and throughput on the BIOSCAN-TRAITS workload using two NVIDIA H100 80GB GPUs. The SAE introduces only a small overhead: DINOv2 activation computation and the SAE forward pass together take

Table 4: Runtime and throughput of the proposed pipeline, measured on two NVIDIA H100 80GB GPUs. Times are averaged over the BIOSCAN-TRAITS workload.

| Task | Time | Remarks |
|---|---|---|
| Activation computation (1 image) | 2.74 ms | DINOv2 backbone |
| SAE forward (1 image) | 4.53 ms | Sparse Autoencoder |
| **Total preprocessing (1 image)** | 7.26 ms | Feature extraction + SAE |
| MLLM inference (3 images / annotation) | 4.62 s | Qwen2.5-VL-72B |
| Throughput (2 NVIDIA H100 80GB GPUs) | 208.9 annotations/h/GPU | |

7.26 ms per image, whereas MLLM inference (conditioning on three SAE-selected patches per image) dominates the budget at 4.62 s per annotation. A cost-of-use analysis comparing public API pricing (Qwen2.5-VL-72B vs. GPT-5-mini) is provided in Appendix H.

## 4.6 FINE-TUNING WITH TRAIT SUPERVISION

Table 5: Zero-shot species classification accuracy (%) on the Insects (Ullah et al., 2022) benchmark. Incorporating trait-level supervision yields clear gains over the baseline pretrained model. BioCLIP 2 is pretrained on BIOSCAN-5M; therefore, we evaluate it directly under trait-level supervision.

| Model | BioCLIP | BioCLIP 2 |
|---|---|---|
| Baseline | 34.8 | 55.3 |
| Baseline + species-level fine-tuning (BIOSCAN-TRAITS) | 39.6 | – |
| Baseline + trait-level fine-tuning (BIOSCAN-TRAITS) | **39.9** | **56.23** |

To assess the utility of our morphological trait description dataset, we fine-tuned BioCLIP (Stevens et al., 2024; Gu et al., 2025), a biologically grounded vision–language foundation model on this dataset. When evaluated on Insects (Ullah et al., 2022), a volunteer-labeled, in-the-wild benchmark, this yielded a significant gain in zero-shot species classification over the pre-trained model (Table 5). This provides initial evidence that trait-level supervision supports better generalization, underscoring the potential of our dataset for training biologically grounded foundation models.

Notably, sparse autoencoders disentangle foreground from background by activating distinct neuron subsets. By aggregating consistent traits across multiple images per species, our pipeline further improves robustness to real-world noise. As a result, models fine-tuned on SAE-derived trait descriptions generalize more effectively to challenging, in-the-wild imagery.

## 5 CONCLUSION

We present a novel pipeline for distilling morphological traits into high-fidelity, natural language descriptions by leveraging sparse autoencoders and multimodal language models. Applied to the BIOSCAN-5M dataset, our method produces a large-scale corpus of over 80K trait descriptions across 19K insect images, constituting one of the first datasets to provide structured, interpretable trait-level supervision at scale. BIOSCAN-TRAITS can support ecology applications such as scaling trait databases and enabling morphology–environment analyses from existing image repositories. Through extensive analysis, we examine the impact of key design factors, including the use of multiple images for trait verbalization, trait frequency thresholds, sparsity levels in the autoencoder, and the choice of MLLM backbone, on the precision and accuracy of generated traits. Integrating trait-level supervision improves generalization in downstream tasks such as fine-grained species classification, underscoring the utility of our proposed pipeline-generated datasets for biologically grounded learning. We discuss the limitations of our approach in Appendix A. Looking forward, we aim to extend this pipeline to construct large-scale datasets across diverse biological domains and across multiple taxonomic levels, enabling domain-specific vision-language models with improved robustness, interpretability, and ecological relevance for large-scale biodiversity applications.

ETHICS STATEMENT

This work advances global biodiversity conservation by introducing a scalable trait annotation pipeline for generating image-to-trait datasets, which can support the development of biologically grounded foundation models. Such models have the potential to improve species recognition, facilitate understanding of evolutionary patterns, and inform conservation strategies in the context of climate change. By reducing reliance on expert-curated annotations, our approach democratizes access to morphological data and empowers under-resourced institutions and citizen science efforts with automated analysis tools. However, errors in trait interpretation, such as those arising from hallucination or domain shift, may propagate into downstream applications, including species classification and conservation decision-making. It is therefore essential that these tools be deployed in close collaboration with domain experts to ensure reliability and accuracy.

REPRODUCIBILITY STATEMENT

Our code is available at `github.com/OSU-NLP-Group/sae-trait-annotation`, and the dataset can be accessed at `huggingface.co/datasets/osunlp/bioscan-traits`. The hyperparameter settings are listed in Table D.4, the trait generation pipeline is outlined in Algorithm 1, and prompt templates are provided in Appendix C. All experiments were performed on NVIDIA H100 GPUs.

ACKNOWLEDGEMENTS

We thank colleagues in the OSU NLP group for valuable feedback. This research was supported in part by NSF CAREER #2443149, NSF OAC 2118240, and an Alfred P. Sloan Foundation Fellowship. We also acknowledge computational resources provided by the Ohio Supercomputer Center Ohio Supercomputer Center (1987). This work was in part conceived at Funcapalooza.[2] S. Record and A. East were additionally supported by the US National Science Foundation's Award No. 242918 (EPSCOR Research Fellows: NSF: Advancing National Ecological Observatory Network-Enabled Science and Workforce Development at the University of Maine with Artificial Intelligence) and by Hatch project Award #MEO-022425 from the US Department of Agriculture's National Institute of Food and Agriculture.

---

[2]Website: `github.com/Imageomics/FuncaPalooza-2025/wiki/`.

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

APPENDICES

This supplementary material provides additional details omitted in the main text.

# Contents

## A    LIMITATIONS

We assume that the dense features from the backbone image foundation model encode morphology-relevant signals. If these representations are biased toward generic visual concepts, important biological traits may be underrepresented. The SAE discovers latent factors that are spatially and semantically coherent, but some latents might correspond to multiple co-occurring traits (*e.g.*, "elongated + thin"). This can make it difficult to disentangle fine-grained trait attributes or compositional traits. Trait descriptions generated with smaller MLLMs like Qwen-2.5-VL-7B are susceptible to hallucination, particularly when prompted with noisy or background-dominated patches. Also, evaluating trait correctness at scale remains a challenge due to the absence of ground-truth morphological trait annotations.

Recent work (Kantamneni et al., 2025; Wu et al., 2025) has highlighted the limitations of SAEs, showing that they do not consistently outperform simpler baselines on downstream tasks. However, we do not use SAEs for steering or sparse probing in LLMs, but rather as a pragmatic tool for proposing spatially localized, candidate part detectors in DINOv2 features that can be grounded to image patches and then described by an MLLM. We mitigate some known SAE limitations by (i) applying species-contrastive ranking and frequency thresholds to filter out spurious latents, (ii) enforcing multi-image consistency (traits must recur across many instances of the same species), and (iii) evaluating the resulting traits both via expert ratings and via downstream transfer to in-the-wild Insects classification. In other words, we do not assume that SAE features are the true underlying traits; instead, we treat them as a useful decomposition that is subsequently empirically validated and filtered.

## B    COMPREHENSIVE RESULTS

The comprehensive results with standard deviation for ratings for various ablations are given in Table B.1, Table B.2, and Table B.3, respectively.

Table B.1: Incorporating latent-specific patches significantly improves the quality of trait descriptions. Including multiple images in the prompt encourages MLLMs to focus on the traits common across all images, at the cost of more tokens per query. Using multiple images with SAE-extracted bounding boxes leads to improved precision, as better ratings indicate. The experimental setup uses Qwen2.5-VL-72B as MLLM, a normalized frequency threshold of $(t_{\text{freq}}) = 3e-3$, and 1,000 input images.

| Method | # Images | # Tokens | # Images | # Traits | Avg. Rating |
|---|---|---|---|---|---|
| MLLM | 1 | 413 | – | – | 3.00 (±0.71) |
| MLLM | 3 | 940 | – | – | 3.15 (±0.54) |
| MLLM + SAE | 1 | 411 | 460 | 9,435 | 3.84 (±0.63) |
| MLLM + SAE | 3 | 1,072 | 460 | 7,897 | **3.91** (±0.92) |

Table B.2: SAEs often trade off between reconstruction error (MSE) and sparsity ($L_0$). We investigate the effect of choosing between different balances of these errors. We find that lower sparsity performs better for both values of frequency threshold ($t_{\text{freq}}$). A lower value of the sparsity coefficient ($\alpha$) leads to lower MSE and thus better reconstruction. It improves the coverage of latents, leading to better recall. The experimental setup uses an input dataset of 1,000 images.

| Method | $\alpha$ | $t_{\text{freq}}$ | SAE MSE | SAE $L_0$ | # Images | # Traits | Avg. Rating |
|---|---|---|---|---|---|---|---|
| MLLM+SAE | 2e−4 | 1e−2 | 8.8e−3 | 1,081.1 | 60 | 60 | 3.84 (±0.70) |
| MLLM+SAE | 4e−4 | 3e−3 | 2.7e−2 | 690.4 | 460 | 7,897 | **3.91** (±0.92) |
| MLLM+SAE | 4e−4 | 1e−2 | 2.7e−2 | 690.4 | 20 | 20 | 3.58 (±1.05) |
| MLLM+SAE | 8e−4 | 3e−3 | 5.4e−2 | 242.2 | 458 | 3,060 | 3.87 (±0.83) |

**MLLM Quality Ablations.** To evaluate the impact of model scale on morphological trait generation, we compare descriptions produced by Qwen2.5-VL-7B and Qwen2.5-VL-72B (Wang et al., 2024)

Table B.3: We investigate the effect of the verbalizer MLLM for morphological trait extraction for both the MLLM-only and MLLM + SAE models. We observe that GPT-5-mini achieves the highest average rating, outperforming both open Qwen-2.5-VL variants by a substantial margin. The larger Qwen-2.5-VL-72B model (Wang et al., 2024) consistently obtains better ratings than its 7B counterpart. We note that GPT-5 mini and Qwen-2.5-VL-7B models might lead to false positives due to hallucination while extracting common traits in three input SAE-annotated images (Figure B.1). In contrast, the Qwen2.5-VL-72B model demonstrates improved robustness, avoiding such hallucinations and yielding more accurate trait descriptions. The experimental setup uses an input dataset of 20K images and $t_{\text{freq}} = 1\mathrm{e}{-2}$.

| Method | MLLM | # Images | # Traits | Avg. Rating |
|---|---|---|---|---|
| MLLM | Qwen-2.5 VL 7B | 476 | 536 | 2.85 (±0.67) |
| MLLM | Qwen-2.5 VL 72B | 371 | 411 | 3.15 (±0.54) |
| MLLM + SAE | Qwen-2.5 VL 7B | 478 | 538 | 2.90 (±1.39) |
| MLLM + SAE | Qwen-2.5 VL 72B | 358 | 370 | 3.58 (±1.05) |
| MLLM + SAE | GPT-5 mini | 478 | 538 | **4.04** (±0.45) |

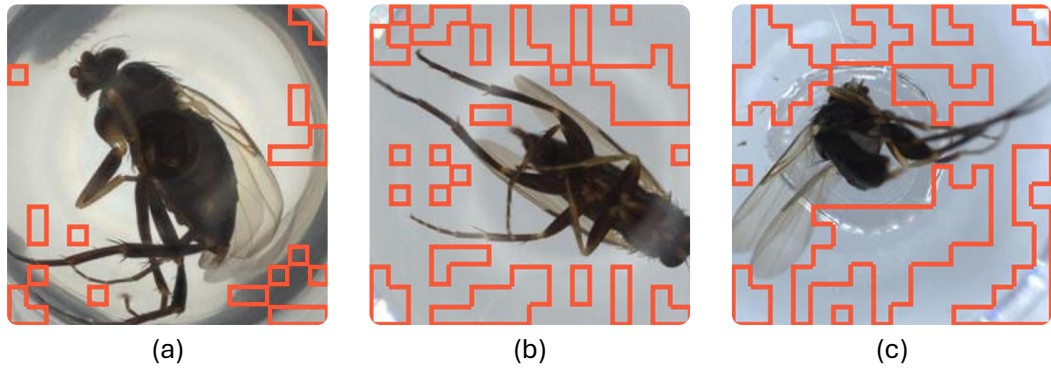

(a)        (b)        (c)

**Qwen2.5-VL-7B**: Wing: Transparent, delicate, slightly curved, light brownish color, fine texture.
**GPT-5 mini:** - Wing: translucent, pale, elongated, Legs: dark, slender, jointed, - Body (thorax/abdomen): dark brown, rounded, glossy.
**Qwen2.5-VL-72B**: None: No shared visible traits across all three regions.

Figure B.1: Comparison of morphological trait description quality between Qwen2.5-VL-7B, GPT-5 mini, and Qwen2.5-VL-72B for *Diplonevra nitidula*. Each red box highlights a region selected by SAE neurons with high activation, indicating regions used for prompting the MLLM + SAE. The Qwen2.5-VL-72B model correctly recognizes the background context and refrains from hallucinating visible traits, suggesting improved spatial grounding.

when prompted with latent-indexed image patches (Table B.3). The larger 72B model consistently receives higher human evaluation scores than its 7B counterpart. In one illustrative example, Qwen2.5-VL-72B correctly identifies a red-boxed region as background, while the 7B model incorrectly hallucinates a body part description (Figure B.1). These results suggest that larger models exhibit improved spatial grounding and are more reliable in avoiding false positive trait attributions.

## C    System Prompts

The prompts used for the MLLM + SAE model are shown in Figure C.2 and Figure C.3, corresponding to the multi-image and single-image settings, respectively. For comparison, the prompts for the MLLM-only baseline are provided in Figure C.4 (multi-image) and Figure C.5 (single-image).

You are given three images of insects, each with multiple red bounding boxes highlighting specific regions.

For each image:
1. For every highlighted region, determine whether it contains a visible insect body part or just background. If it is mostly background, respond with "background".

2. If it contains a visible body part, identify which part it is (*e.g.*, leg, wing, antenna), and describe its visible morphological traits: shape, size, color, texture, and any distinct markings. Use only the visual information present in the image.
After analyzing all three highlighted regions in images:

3. Identify and list the morphological traits that are **common across all three regions**, **solely based on what is visible in all images**.

**Important Instructions**:
- Do not infer or assume information that is not directly observable. Avoid adding external knowledge.
- Use only what is clearly visible.
- Be concise. Limit the total response to under 200 tokens.

**Output Format**:
- [Image 1]:
- **[Body Part]:** [Visible trait]
- [Image 2]:
- **[Body Part]:** [Visible trait]
- [Image 3]:
- **[Body Part]:** [Visible trait]
- [Common Traits Across All Three Images]:
- **[Body Part]:** [Shared visible trait]
...

Figure C.2: Prompt for MLLM + SAE (multiple images)

You are given an image of an insect with multiple red bounding boxes overlaid on it, highlighting a specific region.

1. Determine whether the highlighted region contains a visible body part of the insect or only the background. If it appears to be background, respond with "background".

2. If it contains a visible body part, identify which part it is. Then, briefly describe the observable morphological features - such as shape, size, color, texture, or distinct markings - **based solely on what is visible in the image**.

**IMPORTANT**: Do not infer or assume information that is not directly observable. Avoid adding external knowledge.

Figure C.3: Prompt for MLLM + SAE (single image)

You are given three images of insects. Your task is to visually analyze them and extract observable morphological traits.

1. Identify the visible body parts of the insect (e.g., head, thorax, abdomen, legs, wings, antennae), **common in all three images**.
2. For each part, identify its morphological features - such as shape, size, color, texture, or distinct markings.
3. After analyzing all three images individually, list the morphological traits that are **common across all three insects**. **Only output traits that are visibly consistent across all images**.

**IMPORTANT**: Do not infer or assume information that is not directly observable. Avoid adding external knowledge.

Figure C.4: Prompt for MLLM-only baseline (multiple images)

You are given an image of an insect specimen. Your task is to visually examine the insect and describe its observable morphological traits.

1. Identify the visible body parts of the insect (*e.g.*, head, thorax, abdomen, legs, wings, antennae).
2. For each part, briefly describe the observable morphological features - such as shape, size, color, texture, or distinct markings - **based solely on what is visible in the image**.

**IMPORTANT**:
1. Do not infer or assume information that is not directly observable. Avoid adding external knowledge.
2. Keep your response concise and under 200 tokens.

Figure C.5: Prompt for MLLM-only baseline (single image)

# D  EXPERIMENTAL SETUP

## D.1  HYPERPARAMETER CONFIGURATION

Table D.4 summarizes all hyperparameters used for SAE training and dataset generation. We experiment with different learning rate values and choose $1e-3$ based on qualitative inspection of learned traits. All experiments were conducted on NVIDIA H100 GPUs. SAE training required approximately 11 hours, while the dataset generation took 193 hours using a single process on 2 GPUs.

## D.2  DOWNSTREAM EVALUATION

For downstream evaluation, we use the Insects dataset (Ullah et al., 2022), which consists of volunteer field photos of live insects interacting with flowers and foliage, often partially occluded, in diverse poses, backgrounds, and viewing distances. This introduces multiple distribution shifts (background clutter, illumination, pose, occlusion, and scale) beyond the lab setting. We fine-tune BioCLIP in a standard image–text contrastive manner, where the text input is a caption that concatenates the species name with the trait description. Concretely, we use prompts of the form "A photo of <species-name> with <trait-description>."

# E  FEATURE DETECTOR ABLATIONS

We use DINOv2-base (ViT-B/14) (Oquab et al., 2024) as our feature extractor, motivated by prior work showing its effectiveness in producing high-quality SAE representations (Stevens et al., 2025; Pach et al., 2025). To validate this choice, we conducted preliminary experiments on a 1000-species benchmark derived from BIOSCAN-5M (20 train / 30 test images per species), comparing CLIP ViT-B/16 (Caron et al., 2021) and DINOv2-base features (Table E.6).

Table D.4: Hyperparameters for SAE training, filtering, and downstream fine-tuning.

| Hyperparameter | Value |
|---|---|
| Hidden Width | 24,576 (32× expansion) |
| Sparsity Coefficient $\alpha$ | {2e−4, 4e−4, 8e−4} |
| Sparsity Coefficient Warmup | 500 steps |
| Batch Size | 16,384 |
| Learning Rate $\eta$ | {5e−4, 1e−3} |
| Learning Rate Warmup | 500 steps |
| Activation threshold $t_{activation}$ | 0.9 |
| ViT layer ID | 10 |
| BioCLIP FT Learning Rate | 3e−4 |
| BioCLIP FT Warmup | 2,000 steps |
| BioCLIP 2 FT Learning Rate | 2e−5 |
| BioCLIP 2 FT Warmup | 2,000 steps |

Table D.5: Dataset statistics. On average, each image is associated with 4.2 trait samples.

| Metric | Value |
|---|---|
| # Species | 736 |
| # Genera | 417 |
| # Unique images | 19.1K |
| # Samples | 80.8K |

We observed that DINOv2-base substantially outperforms CLIP ViT-B/16, using the kNN classifier. Based on these results, we selected DINOv2-base as our backbone. Following prior work (Stevens et al., 2025), we extract features from the penultimate layer of the ViT for SAE training.

Table E.6: Species classification accuracy on 1000-species benchmark derived from BIOSCAN-5M (20 train / 30 test images per species). DINOv2-base substantially outperforms CLIP ViT-B/16, using the kNN classifier.

| Model | Top1 Accuracy (%) |
|---|---|
| CLIP ViT-B/16, $k$NN | 24.57 |
| SigCLIP ViT-B/16, $k$NN | 29.68 |
| DINOv2-base, $k$NN | **41.28** |

## F CROWDSOURCING DETAILS

All trait description ratings were performed solely by the authors of this paper, who voluntarily participated in the evaluation. The IRB indicated that our research is exempt and does not require approval. The evaluation rubric is shown in Table F.7.

## G DATASET EXAMPLES

We use the BIOSCAN-5M (Gharaee et al., 2024) for training the SAE models and for dataset generation. It is licensed under the Creative Commons Attribution 3.0 Unported license, which permits its use for academic research. Trait annotation examples from BIOSCAN-TRAITS are shown in Figures G.6–G.10.

## H ANNOTATION COST ANALYSIS

Table H.8 summarizes the cost-of-use when calling Qwen2.5-VL-72B and GPT-5-mini via public APIs. Closed models such as GPT-5-mini offer stronger performance at lower marginal API cost. In

Table F.7: Example-based rubric for evaluating trait descriptions.

| Score | Example Image | Evaluation Criteria |
|---|---|---|
| 5 | 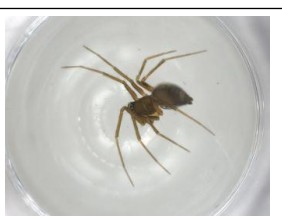 | **Completely Correct** — Body part correctly identified; all traits visibly match (color, texture, shape, size); no hallucinations.
Example: "[Leg]: Thin, elongated, light brown, segmented." |
| 4 | 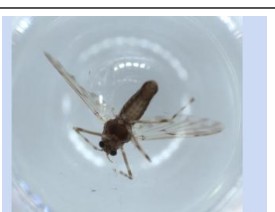 | **Mostly Correct** — Body part is correct; Most traits are accurate, one minor imprecision.
Example: "Leg: Thin, segmented, translucent; jointed structure" (leg is not translucent, but other traits are correct) |
| 3 | 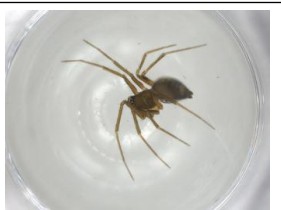 | **Partially Correct** — Body part correct; 1–2 traits vague or incorrect.
Example: "[Leg]: Thick, black, elongated." (body part is correct, but leg is thin and brown.) |
| 2 | 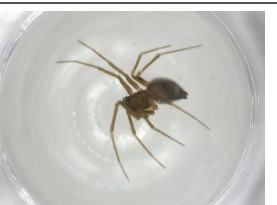 | **Mostly Incorrect** — Incorrect body part or major trait mismatches.
Example: "[Segmented]: All parts are visibly segmented" (the body part is missing, the segmented trait is correct though) |
| 1 | 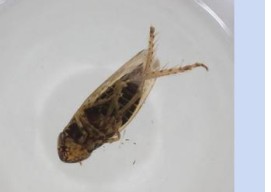 | **Completely Incorrect** — Hallucinated or wrong body part.
Example: "[Antennae]: dark brown." (No antennae are visible) |

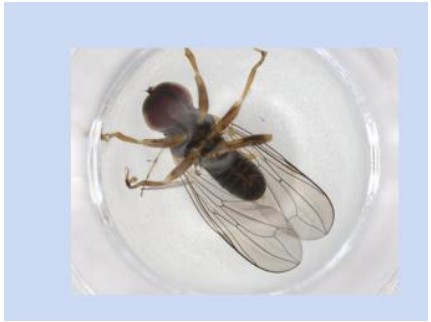

Figure G.6: Example 1 from BIOSCAN-TRAITS: "- Wing: Transparent, elongated, with visible veins. - Antenna: Thin, segmented, light brown".

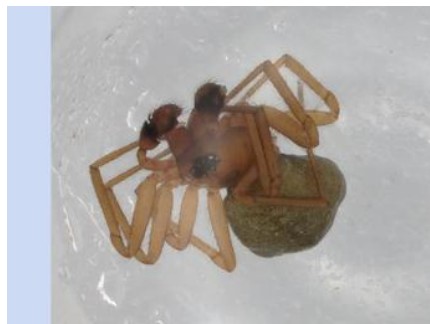

Figure G.7: Example 2 from BIOSCAN-TRAITS: "- [Leg]: Thin, elongated, light brown, segmented".

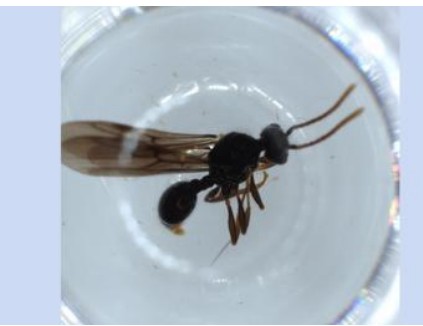

Figure G.8: Example 3 from BIOSCAN-TRAITS: "- Wing: Transparent, elongated, with visible veins. - Antenna: Thin, segmented, dark brown".

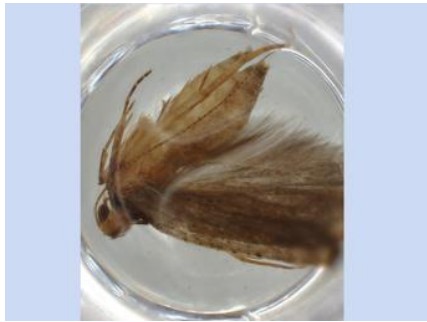

Figure G.9: Example 4 from BIOSCAN-TRAITS: "- Wing: Brown, translucent, folded, with visible veins".

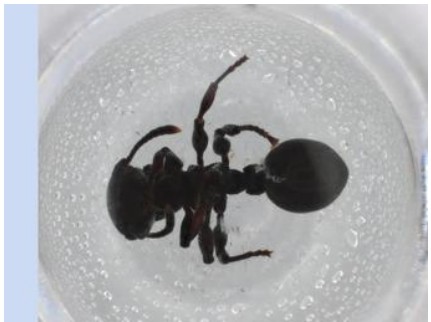

Figure G.10: Example 5 from BIOSCAN-TRAITS: "- Antenna: Thin, elongated, segmented, dark color".

Table H.8: Cost-of-use analysis for generating trait annotations with Qwen2.5-VL-72B (together.ai) and GPT-5-mini APIs, reported as average cost per annotation and extrapolated total cost for processing 100K images. The cost is averaged over the BIOSCAN-TRAITS workload, with 1,072 input tokens and 250 output tokens per annotation.

| Model | Cost per annotation | Approx. cost for 100K images |
|---|---|---|
| Qwen2.5-VL-72B API (together.ai) | $4.1\times10^{-3}$/annotation | $410 |
| GPT-5-mini API | $8\times10^{-4}$/annotation | $80 |

practice, the open-source Qwen2.5-VL-72B can be hosted in-house, shifting cost from per-call API pricing to amortized compute, and allowing users with data-governance constraints to keep images on-premises.

## I    ECOLOGY APPLICATIONS

Below, we outline several concrete ways in which ecologists can leverage the proposed trait-generation pipeline:

- **Expanding trait databases**: Building trait databases by hand using domain experts is time-consuming. An automated tool can quickly add thousands of traits from existing images, populating databases or filling gaps. This helps ecologists who rely on traits (for example, to model species' niches or ecosystem roles) by providing many more data points.
- **Enabling new analyses**: With rich trait labels attached to images, researchers can study correlations between morphology and environment or behavior at scale. For instance, you could analyze how wing shapes vary across climates, or link body color patterns to predation risk. Traits explain ecological patterns better than just species names, and an automated pipeline makes these analyses feasible on large collections.
- **Boosting identification tools**: As shown with BioCLIP, trait-annotated images can improve automatic species-identification models. Models trained on trait captions learn more nuanced visual cues, making them more robust to new specimens or image conditions.

Overall, our pipeline provides a scalable way to inject expert-like knowledge (descriptions of body parts) into machine learning without manual annotation. By turning images into meaningful trait statements, it bridges the gap between digitized specimens and quantitative trait databases, supporting a wide range of biodiversity and ecological research.

## J    ADDITIONAL NEURON ACTIVATION ANALYSIS

Similar to Section 4.3, we analyze additional cases of the top-activating neurons (or latent dimensions) in the SAE to investigate whether they correspond to meaningful morphological traits (Figure J.11). For instance, we observe that within the SAE, neuron 4040 consistently activates on the thorax, while

neuron 16584 responds to the leg-body junction, highlighting spatially grounded morphological regions.

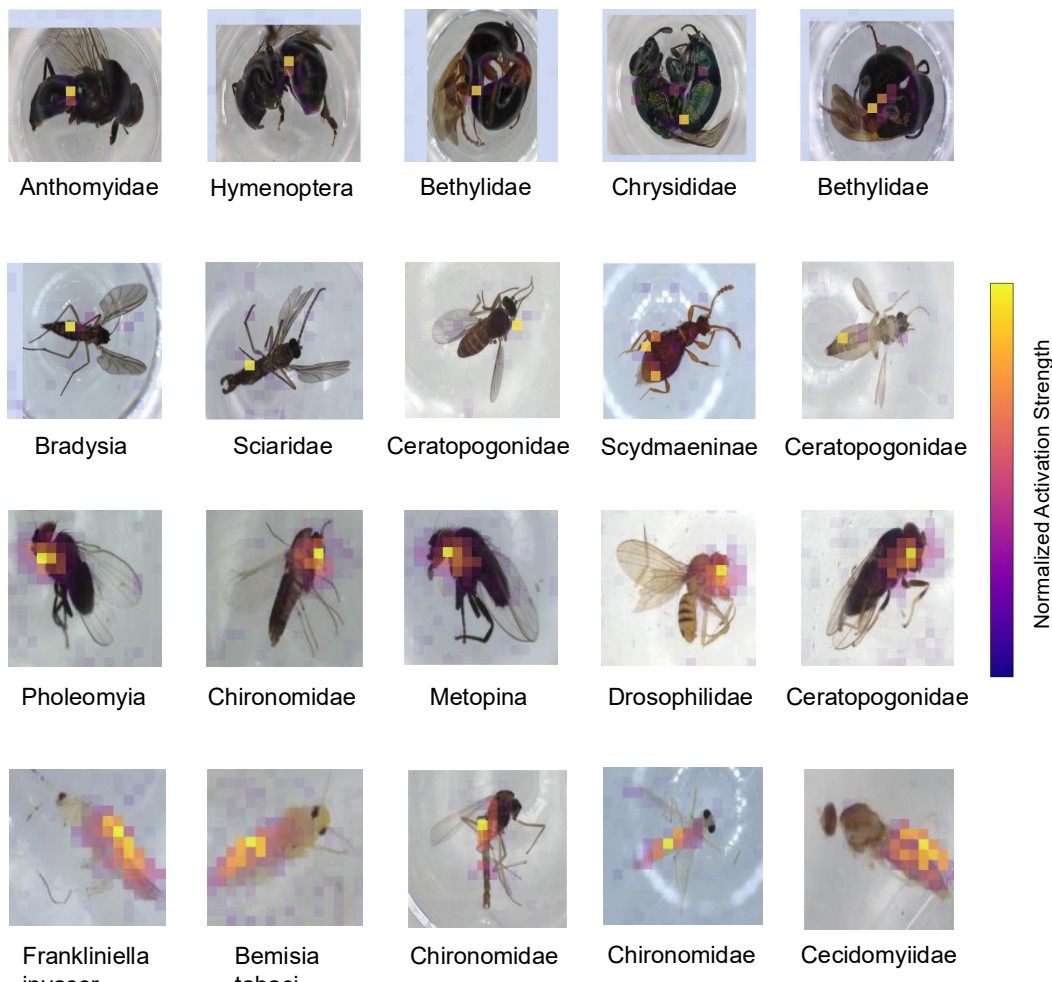

Figure J.11: Neurons 4040, 16584, 13433, and 14153 in SAE get activated at the thorax, at the place where the leg attaches to the body, eyes, and the abdomen, respectively. The labels denote the highest annotated taxonomic level.

## K    ADDITIONAL DATASET ABLATION EXAMPLES

### K.1    MLLM + SAE VS. MLLM-ONLY BASELINE

Figure K.12-K.14 present additional examples comparing the salient morphological trait descriptions generated by the MLLM-only baseline versus MLLM + SAE.

### K.2    QWEN-2.5-VL-7B VS. QWEN-2.5-VL-72B

Figure K.15-K.16 present additional examples comparing the salient morphological trait descriptions generated by Qwen-2.5-VL-7B vs. Qwen-2.5-VL-72B as the backbone MLLM for MLLM + SAE. The larger Qwen2.5-VL-72B model accurately identifies the insect's body parts and avoids the hallucinations observed in its 7B counterpart.

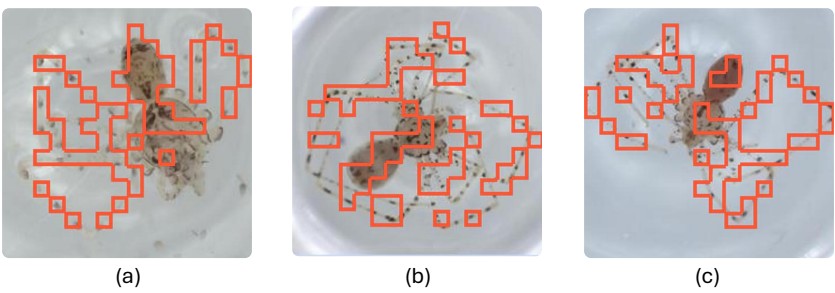

**MLLM**: Common Morphological Traits Across All Three Images: 1. Head: - *Shape*: Rounded, slightly elongated.- *Color/Texture*: Light-colored with darker speckled patterns.- *Distinct Markings*: Dark spots or patches scattered across the surface. 2. Thorax: - *Shape*: Small and compact, merging seamlessly with the head. - *Color/Texture*: Similar light coloration as the head, with dark speckles. - *Distinct Markings*: Speckled pattern continues from the head. 3. Abdomen: - *Shape*: Oval-shaped, larger than the head and thorax combined. - *Color/Texture*: Light base color with a dense covering of dark speckles. - *Distinct Markings*: Dark speckles form a mottled appearance.
**MLLM + SAE**: **[Thin, elongated structure]**: Segmented appearance, brownish/dark coloration.

Figure K.12: Comparison of salient morphological trait description generation using a just MLLM vs. MLLM + SAE for *Scytodes intricata*. Each red box highlights a region selected by SAE neurons with high activation, indicating regions used for prompting the MLLM + SAE.

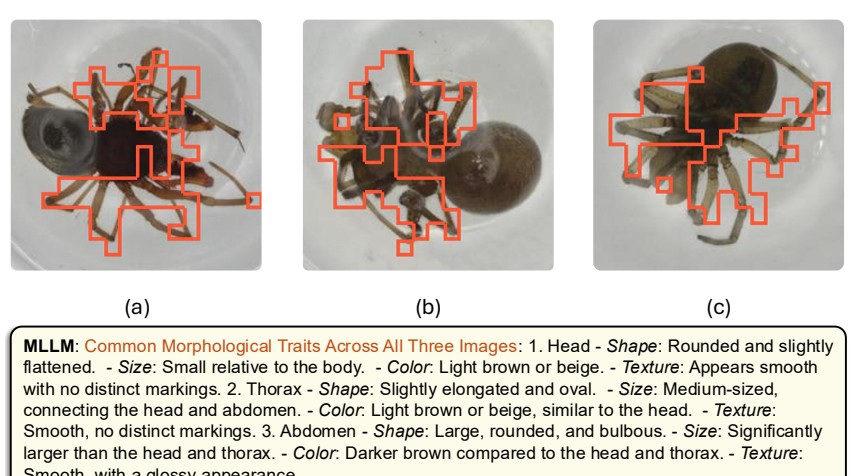

**MLLM**: Common Morphological Traits Across All Three Images: 1. Head - *Shape*: Rounded and slightly flattened. - *Size*: Small relative to the body. - *Color*: Light brown or beige. - *Texture*: Appears smooth with no distinct markings. 2. Thorax - *Shape*: Slightly elongated and oval. - *Size*: Medium-sized, connecting the head and abdomen. - *Color*: Light brown or beige, similar to the head. - *Texture*: Smooth, no distinct markings. 3. Abdomen - *Shape*: Large, rounded, and bulbous. - *Size*: Significantly larger than the head and thorax. - *Color*: Darker brown compared to the head and thorax. - *Texture*: Smooth, with a glossy appearance.
**MLLM + SAE**: **[Leg]**: Thin, elongated, brown, segmented.

Figure K.13: Comparison of salient morphological trait description generation using a just MLLM vs. MLLM + SAE for *Erigone psychrophila*. Each red box highlights a region selected by SAE neurons with high activation, indicating regions used for prompting the MLLM + SAE.

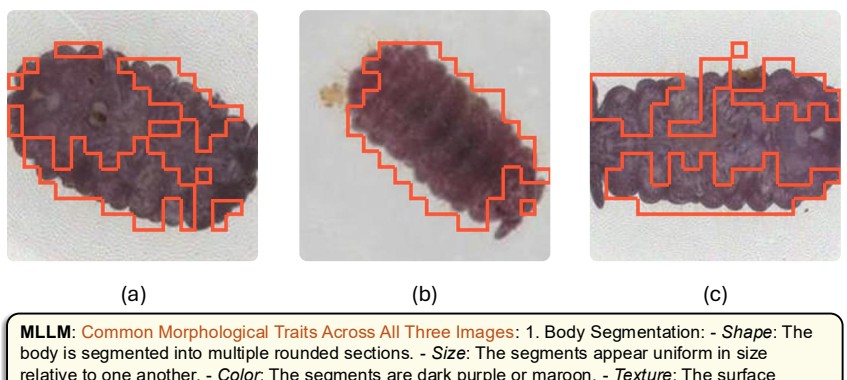

(a)        (b)        (c)

**MLLM**: Common Morphological Traits Across All Three Images: 1. Body Segmentation: - *Shape*: The body is segmented into multiple rounded sections. - *Size*: The segments appear uniform in size relative to one another. - *Color*: The segments are dark purple or maroon. - *Texture*: The surface appears rough with small protrusions or bumps on each segment. 2. Head: - *Shape*: The head is slightly smaller than the other segments and is located at one end of the body. - *Color*: The head matches the dark purple or maroon color of the rest of the body. - *Distinct Markings*: A lighter-colored area is visible near the front of the head, possibly indicating sensory organs or mouthparts. 3. Antennae: - *Shape*: Two short, thin antennae are visible extending from the head.
**MLLM + SAE: Body**: Dark brown, textured surface; irregular shape.

Figure K.14: Comparison of salient morphological trait description generation using a just MLLM vs. MLLM + SAE for *Morulina thulensis*. Each red box highlights a region selected by SAE neurons with high activation, indicating regions used for prompting the MLLM + SAE.

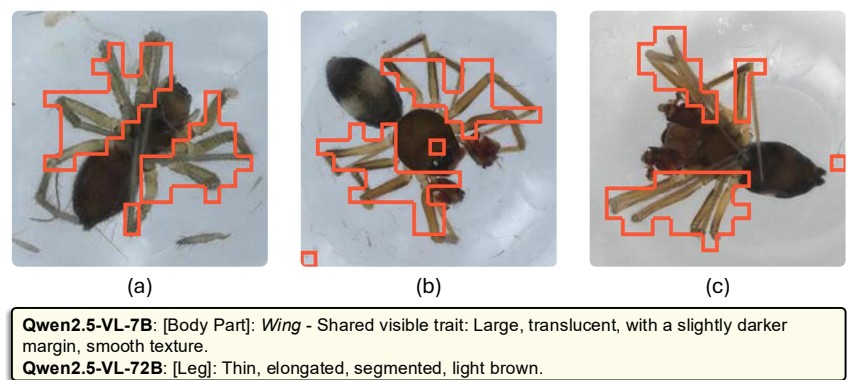

(a)        (b)        (c)

**Qwen2.5-VL-7B**: [Body Part]: *Wing* - Shared visible trait: Large, translucent, with a slightly darker margin, smooth texture.
**Qwen2.5-VL-72B**: [Leg]: Thin, elongated, segmented, light brown.

Figure K.15: Comparison of morphological trait description quality between Qwen2.5-VL-7B and Qwen2.5-VL-72B for *Agyneta straminicola*. Each red box highlights a region selected by SAE neurons with high activation, indicating regions used for prompting the MLLM + SAE. The larger model correctly identifies the highlighted body part of the insect.

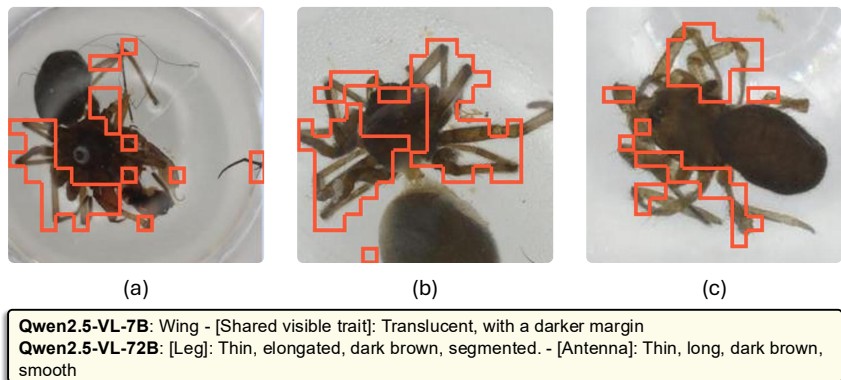

(a)                (b)                (c)

> **Qwen2.5-VL-7B**: Wing - [Shared visible trait]: Translucent, with a darker margin
> **Qwen2.5-VL-72B**: [Leg]: Thin, elongated, dark brown, segmented. - [Antenna]: Thin, long, dark brown, smooth

Figure K.16: Comparison of morphological trait description quality between Qwen2.5-VL-7B and Qwen2.5-VL-72B for *Erigone psychrophila*. Each red box highlights a region selected by SAE neurons with high activation, indicating regions used for prompting the MLLM + SAE. The larger model correctly identifies the highlighted body part of the insect.

### K.3 MULTIPLE VS. SINGLE IMAGE PER LATENT

Figure K.17-K.18 present additional examples comparing the salient morphological trait descriptions generated using a single image versus multiple images for MLLM + SAE. This consensus-driven trait extraction encourages the model to focus on consistent traits and leads to improved precision.

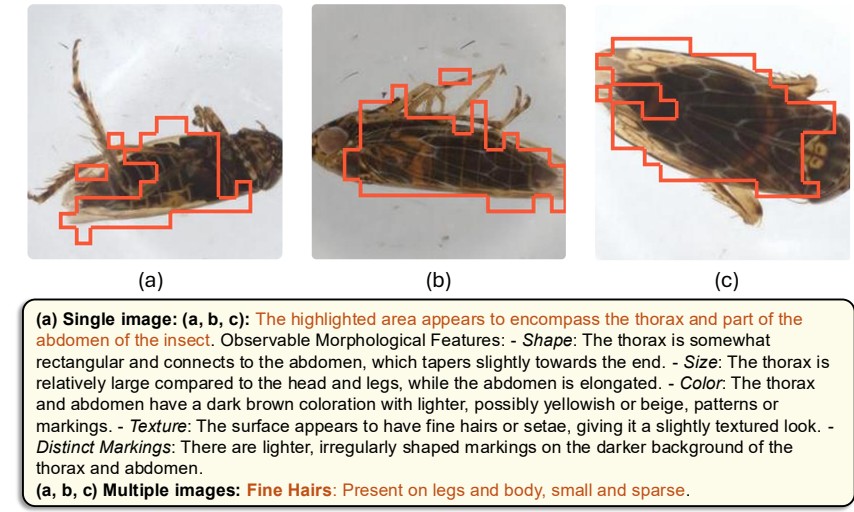

(a)                (b)                (c)

> **(a) Single image: (a, b, c):** The highlighted area appears to encompass the thorax and part of the abdomen of the insect. Observable Morphological Features: - *Shape*: The thorax is somewhat rectangular and connects to the abdomen, which tapers slightly towards the end. - *Size*: The thorax is relatively large compared to the head and legs, while the abdomen is elongated. - *Color*: The thorax and abdomen have a dark brown coloration with lighter, possibly yellowish or beige, patterns or markings. - *Texture*: The surface appears to have fine hairs or setae, giving it a slightly textured look. - *Distinct Markings*: There are lighter, irregularly shaped markings on the darker background of the thorax and abdomen.
> **(a, b, c) Multiple images:** **Fine Hairs**: Present on legs and body, small and sparse.

Figure K.17: Comparison of salient morphological trait description generation using a single image vs. three images for *Deltocephalus fuscinervosus*. Each red box highlights a region selected by SAE neurons with high activation, indicating regions used for prompting the MLLM + SAE. Using multiple images with SAE-extracted bounding boxes leads to dramatically improved precision.

## L  LLM USAGE DETAILS

We utilized large language models (LLMs) to aid in the writing and editing of this paper. Their role within our trait-generation pipeline, specifically the use of multimodal LLMs (MLLMs), is described in Section 3.

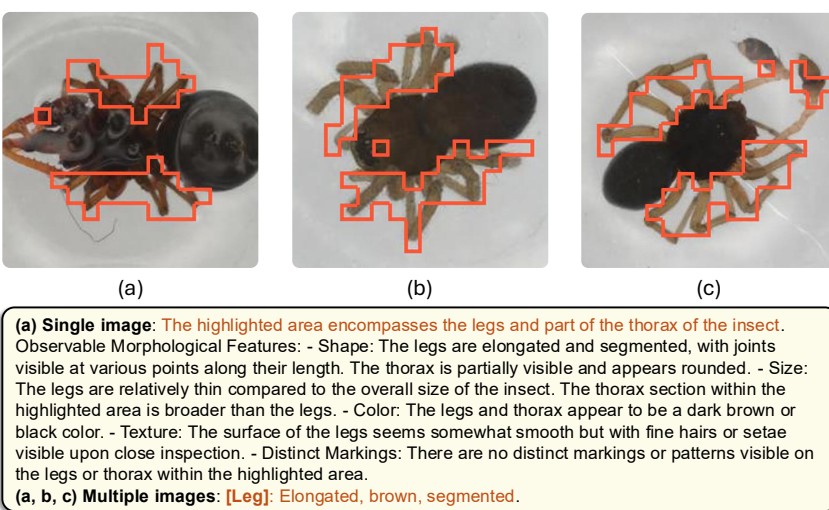

(a)            (b)            (c)

**(a) Single image**: The highlighted area encompasses the legs and part of the thorax of the insect. Observable Morphological Features: - Shape: The legs are elongated and segmented, with joints visible at various points along their length. The thorax is partially visible and appears rounded. - Size: The legs are relatively thin compared to the overall size of the insect. The thorax section within the highlighted area is broader than the legs. - Color: The legs and thorax appear to be a dark brown or black color. - Texture: The surface of the legs seems somewhat smooth but with fine hairs or setae visible upon close inspection. - Distinct Markings: There are no distinct markings or patterns visible on the legs or thorax within the highlighted area.
**(a, b, c) Multiple images**: **[Leg]**: Elongated, brown, segmented.

Figure K.18: Comparison of salient morphological trait description generation using a single image vs. three images for *Erigone arctophylacis*. Each red box highlights a region selected by SAE neurons with high activation, indicating regions used for prompting the MLLM + SAE. Using multiple images with SAE-extracted bounding boxes leads to dramatically improved precision.

