# OpenReview forum: "Automatic Image-Level Morphological Trait Annotation for Organismal Images"
_ICLR.cc/2026/Conference — ICLR 2026 Poster_

### Official Review · Reviewer_nY1L · 2025-10-17

**Soundness:** 2
**Presentation:** 4
**Contribution:** 3
**Rating:** 6
**Confidence:** 4

**Summary:**

Morphological traits (the measurable physical characteristics of organisms) accurately predict how species interact with their environments, but extracting these traits remains a slow, expert-driven process. This paper contends that sparse autoencoders trained on foundation-model features yield monosemantic, spatially grounded neurons that consistently activate on meaningful morphological parts. Leveraging this property, we introduce a trait annotation pipeline that localizes salient regions and uses vision-language prompting to generate interpretable trait descriptions. Using this approach, we construct BIOSCAN-TRAITS, a dataset of 80K trait annotations spanning 19K insect images from BIOSCAN-5M. Human evaluation confirms the biological plausibility of the generated morphological descriptions. When used to fine-tune BioCLIP, a biologically grounded vision-language model, BIOSCAN-TRAITS improves zero-shot species classification on the in-the-wild Insects benchmark.

Authors' contributions: The recognition that sparse autoencoders (SAEs) can be used as interpretable part-detectors for trait extraction. In practice, training an SAE over pre-trained image features produces units whose activations map back onto tight, spatially coherent regions, and a method taking advantage of SAEs as trait detectors. A new dataset, BIOSCAN-TRAITS, and some experiments showing that it is useful for downstream classification tasks.

**Strengths:**

* This is an unusually well-written paper. I very much appreciate the clear dual expertise of the authors, which allows them to efficiently and accurately describe the mechanism by which SAEs operate as well as the challenges of Morphological Trait Extraction. The Related Works section of this paper, while not long, is good.
* The topic is important, and understudied; I'm very happy to see more work in this area.
* Table 4 is a nice contribution; it's a nice finding that BIOSCAN-TRAITS works better than species-only fine-tuning.

**Weaknesses:**

I do have some concerns about the authors overclaiming particularly in the introduction, and the results not being as comprehensive as I would like in certain places.

* The paper spends a long time on introduction and related work; we only get to the actual experiments on page 4. This is not a stylistic problem, as I recognize this paper probably needs more introduction than most, but it does mean there isn't as much room for actual description of the method and experiments. I think the second half of the introduction could be compacted, especially considering it contains many bold statements which aren't all that well supported.
* The method ablations are unusually weak for an empirical paper; only two backbones considered, DINOV2 and ViT-B-16 CLIP, and no ablations on the choice of SAE. A wider range of self-supervised and contrastive backbones would have been nice to see here, e.g. DINOV1, DINOV3, SigLIP, etc, and the decision to use an SAE is absolutely central to the method and should have been ablated.
* Personally I think the Algorithm 1 is not that important for understanding the work and should go to the appendix; I would rather see more core experiments (or ablations) here and the algorithm in the appendix.
* GradCAM is fine as a (weak) initial baseline; a more realistic baseline would be semi-supervised object detection based on target traits similar to (https://acsess.onlinelibrary.wiley.com/doi/10.1002/ppj2.20107), would also be nice to see how PCA, L1-regularized linear models, and Random projections do. These are extremely cheap to run, you don't even need a GPU.
* The authors write that the procedure is fully unsupervised; this is true only if you don't count the backbone -- DINO uses various kinds of weak and self supervision during pretraining, I think it would be clearer to include a mention of this. Furthermore, species-contrastive ranking sounds very much like it would require supervised / labeled data.
* BIOSCAN-5M is an unusually friendly dataset for this type of mapping because the insects are dead and presented mostly centered in the image, and well lit on plates, and there is always exactly 1 insect per image. It would have been nice to have seen an ablation on how well SAEs work on in-the-wild insect data, but failing that, I think the claim should be softened accordingly.
* SAEs are not nearly as robust as the paper's claims would indicate. Sometimes they work well, but often they're no better than much simpler, older approaches. https://arxiv.org/html/2501.16615v1, https://arxiv.org/html/2505.11756, https://arxiv.org/abs/2501.17148, https://arxiv.org/abs/2502.16681v1. Max Tegmark, Neel Nanda, Christopher Manning, Christopher Potts are all doubters of SAEs at this point. I think this should be mentioned as a significant limitation.

If these concerns can be addressed during the rebuttal, I think the paper is worthy of acceptance.

**Questions:**

* Are the images for the main figures cherry-picked? If so, the authors should say so.
* What are the limitations and drawbacks of the authors' approach? Under what circumstances will it be most effective?

---

> ### Author Response · Authors · 2025-11-21
> **Rebuttal by authors (1/2)**
>
> We thank the reviewer for their insightful comments and for recognizing both the quality of the writing and the significance of the problem we address. Below, we provide detailed responses to the reviewer’s comments and questions.
>
> > **(W2) Backbone and SAE ablations**
>
> We appreciate the reviewer’s concern regarding ablations. Our initial choice of DINOv2-base (ViT-B/14) was motivated by prior work [1] showing that its features yield high-quality sparse codes for SAEs and are well-suited to dense, part-level prediction tasks, whereas CLIP-style models are primarily trained to produce global, image-level features. In response to this comment, we have now added experiments with additional feature backbones (all evaluated with a kNN classifier on the same setup in Appendix I):
>
> | Model                 | Top1 Accuracy (%) |
> |-----------------------|-------------------|
> | CLIP ViT-B/16, kNN    | 24.57             |
> | SigCLIP ViT-B/16, kNN | 29.68             |
> | DINOv2-base, kNN      | 41.28             |
> | DINOv3-base, kNN      | 44.30             |
>
> DINOv2-base substantially outperforms CLIP and SigLIP in this setting, and DINOv3-base is marginally stronger still. We note that DINOv3 was released only recently (mid-Aug 2025), well after the start of this project, systematically re-running all SAE and MLLM experiments on DINOv3 is therefore beyond our current rebuttal-time budget. However, the fact that DINOv2 and DINOv3 both perform strongly suggests that our conclusions are not tied to a single backbone, and we will clarify this in the revised manuscript and add the above table.
>
> **SAE design and hyperparameters**
>
> We agree that the SAE is central to our method. We do ablate key SAE-related design choices that directly control the sparsity and selectivity of the learned units. In particular, Sec. 4.3 reports (i) a sweep over the sparsity coefficient $\alpha$, showing how the reconstruction–sparsity trade-off affects the number of active units and expert-rated trait quality, and (ii) a sweep over the species-frequency threshold $t_{\text{freq}}$, showing how stricter species-contrastive filtering trades coverage for precision.
>
> > **(W1) Introduction section**
>
> We appreciate the reviewer’s comment about the balance between introduction and technical content, and we agree that tightening the front matter can make more room for the method and experiments. We have edited the introduction section in the revised manuscript to this effect. We have also added new citations to support several claims in the introduction.
>
> > **(W4) GradCAM and additional baselines**
>
> We agree that GradCAM is a relatively weak baseline, and that other methods could replace the SAE in our pipeline. Furthermore, we appreciate the reference and will add it to our related work. Cunningham et al. [2] find that SAEs are better than PCA, ICA and random directions for sparse decomposition of transformer-based language model activations.
> Unfortunately, running these comparisons in our setting is non-trivial in terms of compute because trying PCA, ICA and random directions requires re-running the entire trait-generation and human-evaluation pipeline (including MLLM captioning and expert rating) for each alternative representation, which is the dominant cost in our workflow (GPU time and annotator effort), rather than the representation learning itself.
>
> > **(W5) Supervised vs. Unsupervised approach**
>
> We thank the reviewer for raising this important clarification. Our earlier phrasing of the trait generation procedure as “fully unsupervised” was imprecise. DINOv2 is trained using self-supervised learning using a teacher–student self-distillation setup. The SAE itself is trained unsupervised on frozen DINOv2 features. In species-constrastive ranking, we use the activation scores to find latents that fire strongly for a target species but remain almost silent for closely related species. This is supervised at the level of species but does not use any trait-level supervision. We have clarified this description in the introduction section of the revised manuscript.
>
> > **(W6) Use of BIOSCAN-5M for experiments**
>
> While we use BIOSCAN-5M for trait generation in this work, our downstream results in Sec. 4.5 show that traits distilled from BIOSCAN-5M features do transfer to in-the-wild Insects data via BioCLIP fine-tuning, indicating that the learned traits capture signal that is useful beyond the lab domain. Finally, prior work [1] has demonstrated that SAEs can discover meaningful concepts in large, in-the-wild image corpora (e.g., ImageNet).
>
> [1] Stevens, Samuel, et al. "Sparse autoencoders for scientifically rigorous interpretation of vision models." arXiv preprint arXiv:2502.06755 (2025).
>
> [2] Cunningham, Hoagy, et al. "Sparse autoencoders find highly interpretable features in language models." ICLR 2024.

---

> > ### Author Response · Authors · 2025-11-21
> > **Rebuttal by authors (2/2)**
> >
> > > **(W7) Limitations of SAEs**
> >
> > We thank the reviewer for pointing out recent work that highlights limitations of sparse autoencoders. We will explicitly cite these works and tone down any language that might suggest SAEs are universally robust or superior to simpler methods.
> >
> > Our claims in this paper are narrower. We use SAEs not for steering or sparse probing in LLMs, but as a **pragmatic tool for proposing spatially localized candidate part detectors** in DINOv2 features that can be grounded to image patches and then described by an MLLM. We further mitigate some of the known issues by (i) applying **species-contrastive ranking** and frequency thresholds to filter out spurious latents, (ii) enforcing **multi-image consistency** (traits must recur across many instances of the same species), and (iii) **evaluating the resulting traits** both via expert ratings and via downstream transfer to in-the-wild Insects classification. In other words, we do not assume that SAE features are “the true” underlying traits; **we treat them as a useful decomposition that is then empirically validated and filtered**. We have added this discussion in Appendix A.
> >
> > > **(Q1) Images for the main figures**
> >
> > We selected examples that highlight common insect parts (e.g., legs, thorax, wings) that are readily interpretable even for readers without entomological expertise. We include additional randomly sampled examples in Appendix H.
> >
> > > **(Q2) Discussion of Limitations**
> >
> > We discuss the limitations of our approach in Section 6 (now moved to main text). First, we assume that the dense features from the backbone image foundation model encode morphology-relevant signals. If these representations are biased toward generic visual concepts, important biological traits may be underrepresented. Second, while the SAE discovers latent factors that are spatially and semantically coherent, some latents might correspond to multiple co-occurring traits (e.g., elongated + thin). This can make it difficult to disentangle fine-grained trait attributes or compositional traits. Third, trait descriptions generated with smaller MLLMs like Qwen-2.5-VL-7B are susceptible to hallucination, particularly when prompted with noisy or background-dominated patches.

---

> > > ### Comment · Reviewer_nY1L · 2025-11-23
> > >
> > > I thank the authors for a strong and thoughtful rebuttal. W5, W7, Q1, Q2 have been fully addressed by these comments, and I look forward to seeing the improvements in the published work. For W4, I now understand that the request is unrealistic in the rebuttal time frame because of the need to rerun the GPU-dependent experiments, so I think adding the citations and  acknowledging this as an important direction for future work would be sufficient. For W2, given the limited time frame, I think this additional ablation is reasonable, and appreciated.

---

### Official Review · Reviewer_npDF · 2025-10-31

**Soundness:** 4
**Presentation:** 3
**Contribution:** 4
**Rating:** 8
**Confidence:** 4

**Summary:**

The paper proposes a pipeline for automatically annotating morphological traits at the image level from organismal images. The method (i) extracts dense visual features (DINOv2) and passes them through a sparse autoencoder (SAE) to obtain monosemantic, spatially grounded latent units; (ii) converts high-activation latents into localized boxes/masks; and (iii) prompts a multimodal LLM (here, Qwen2.5-VL-72B) to produce natural-language trait descriptions for those regions. Applying this to BIOSCAN-5M, the authors release BIOSCAN-TRAITS: ~80k trait annotations over ~19k insect images. Human evaluation indicates these SAE-guided descriptions are more precise than MLLM-only and Grad-CAM baselines. Fine-tuning BioCLIP on BIOSCAN-TRAITS improves zero-shot species classification on the Insects benchmark (+2.9% absolute), suggesting trait-level supervision boosts downstream generalization.

**Strengths:**

This is a well-written paper on a topic of significant importance.

*Clear, modular pipeline with interpretability baked in*: SAEs yield localized, monosemantic units (e.g., “wing”, “antenna”) that ground the text prompts and reduce hallucination. The pipeline is easy to reason about and replicate.

*Substantive dataset contribution*: BIOSCAN-TRAITS (80k traits / 19k images) fills a gap in large-scale, trait-level supervision for biodiversity ML.

*Downstream utility*: Fine-tuning BioCLIP with trait-level labels improves zero-shot accuracy on an in-the-wild benchmark, underscoring practical value.

**Weaknesses:**

*Model coverage of VLMs/LMMs is narrow*: Most experiments rely on Qwen2.5-VL (7B vs. 72B); broader open/closed LMM comparisons (and robustness under domain shift) would strengthen the paper.

*Cost/latency unquantified beyond tokens*: While the paper reports token counts per setting, it lacks a wall-clock + $-cost analysis per annotated trait/image across model sizes and batching regimes. This will be very useful for labs and museums planning large runs.

*BioCLIP only (no BioCLIP2):* Downstream tests fine-tune BioCLIP; given rapid progress in biologically grounded VLMs, results may be stronger (and more current) with BioCLIP2 if available.

*End-user pathways are implicit*. The paper could more concretely describe how curators or ecologists operate the system (inputs, thresholds, QA loop, export formats) and how to integrate trait text with existing collection databases.

**Questions:**

**Why not BioCLIP2?** You fine-tune BioCLIP and show +2.9% on Insects. If BioCLIP2 is available, can you (a) replicate the gain, and (b) test whether trait-level supervision compounds improvements already present in BioCLIP2? Reporting both would isolate the incremental value of trait supervision from backbone/dataset size advances.

**Other LMMs (open vs. closed) and robustness**: Beyond Qwen2.5-VL (7B/72B), have you tried LLaVA-Next, Gemini-Vision, or GPT-4o-mini for trait generation? A compact matrix (quality, hallucination rate, token usage, throughput) would reveal portability, licensing trade-offs, and whether SAE-guided prompting narrows the gap between open and closed models. You already compared Qwen 7B vs. 72B; extending that table to 2–3 additional LMMs would be very informative.

**Cost-of-use analysis:**  Could you add: Per-image cost & time: preprocessing (feature extraction + SAE forward), #patches, tokens, MLLM seconds, and total $ at typical API/on-prem pricing; Throughput under batching and GPU memory constraints for open-weights models. This would let institutions forecast the budget for, say, 100k images.

**How do you envision end-users using it?**  It would help to spell out a reference workflow for curators/biologists
Even a short “Operations” section or a demo video link would greatly increase practical adoption.

---

> ### Author Response · Authors · 2025-11-21
> **Rebuttal by authors (1/2)**
>
> We thank the reviewer for their insightful comments and for recognizing that the paper is well written and that the proposed trait annotation pipeline has good practical utility. Below, we provide detailed responses to the reviewer’s comments and questions.
>
> > **(W1, Q2) Experiments with additional LMMs**
>
> Thanks for the suggestion. Due to time constraints for human rating, we were able to include one additional MLLM (GPT-5-mini) alongside Qwen-2.5-VL-7B and Qwen-2.5-VL-72B. The results are summarized below:
>
> | Method      | MLLM             | # Images | # Traits | Avg. Rating    |
> |------------|------------------|---------:|---------:|----------------|
> | MLLM + SAE | Qwen-2.5 VL 7B   |      478 |      538 | 2.90 (±1.39)   |
> | MLLM + SAE | Qwen-2.5 VL 72B  |      358 |      370 | 3.58 (±1.05)   |
> | MLLM + SAE | GPT-5 mini       |      478 |      538 | **4.04 (±0.45)**   |
>
> We observe that GPT-5-mini + SAE attains the highest average expert rating, outperforming both open Qwen-2.5-VL variants by a substantial margin. However, the GPT-5-mini model is susceptible to false positives due to hallucination while extracting common traits in three input SAE-annotated images, similar to Qwen-2.5-VL-7B (see revised Fig. B.1). For the main experiments in the paper, we therefore retain Qwen-2.5-VL-72B as our primary backbone, as it is open, can be hosted in-house, and is significantly cheaper than closed APIs. We have updated Table B.3 with these results in the revised manuscript.
>
> > **(W2, Q3) Cost of use analysis**
>
> We thank the reviewer for this very practical suggestion. In the revised manuscript, we added a new “Cost and Throughput Analysis” section (Section 4.6) with tables showing the per-image runtime, throughput, and API cost estimates, also shown below:
>
> | **Task**                                  | **Time**        | **Remarks**                 |
> |-------------------------------------------|-----------------|------------------------------------|
> | Activation computation (1 image)          | 2.74 ms         | DINOv2 backbone                    |
> | SAE forward (1 image)                     | 4.53 ms         | Sparse Autoencoder                 |
> | **Total preprocessing (1 image)**         | **7.26 ms**     | **Feature extraction + SAE**       |
> | MLLM inference (3 images / annotation)    | 4.62 s          | Qwen2.5-VL-72B                     |
> | Throughput (2 NVIDIA A100 80GB GPUs)      | 208.9 annotations/h/GPU |        |
>
> *Table: Runtime and throughput of the proposed pipeline, measured on two NVIDIA A100 80GB GPUs. Times are averaged over the Bioscan-Traits workload.*
>
> | **Model**                     | **Cost per annotation ($)**              | **Approx. cost for 100K images ($)** |
> |------------------------------|--------------------------------------|----------------------------------|
> | Qwen2.5-VL-72B API (together.ai) | $4.1 \times 10^{-3}$/annotation      | $410                            |
> | GPT-5-mini API               | $8 \times 10^{-4}$/annotation        | $80                             |
>
> *Table: Cost-of-use analysis for generating trait annotations with Qwen2.5-VL-72B (together.ai) and GPT-5-mini APIs, reported as average cost per annotation and extrapolated total cost for processing 100K images. The cost is averaged over the* Bioscan-Traits *workload, with 1072 input tokens and 250 output tokens per annotation*
>
> > **(W3, Q1) BioCLIP-2 evaluation**
>
> We appreciate the reviewer’s suggestion to consider BioCLIP-2 [1]. BioCLIP-2 is trained on TreeOfLife-200M, which is orders of magnitude larger than Bioscan-Traits (200M vs. 80K images), and already exhibits strong emergent trait prediction capabilities. In contrast, our BioCLIP experiments are designed to isolate the incremental value of trait supervision on top of a strong but not yet saturating backbone.
>
> | Model                     | Accuracy (Insects) (%) |
> | ------------------------- | ---------------------- |
> | BioCLIP-2                 | 55.3                   |
> | BioCLIP-2 + FT on Bioscan-Traits | **56.23**                  |
>
> [1] Gu, Jianyang, et al. "Bioclip 2: Emergent properties from scaling hierarchical contrastive learning." NeurIPS 2025.

---

> > ### Author Response · Authors · 2025-11-21
> > **Rebuttal by authors (2/2)**
> >
> > > **(W4, Q4) How do you envision end-users using it?**
> >
> > Below, we outline several concrete ways in which ecologists can leverage the proposed trait-generation pipeline.
> >
> > - **Expanding trait databases**: Building trait databases by hand using domain experts is time-consuming. An automated tool can quickly add thousands of traits from existing images, populating databases or filling gaps. This helps ecologists who rely on traits (for example, to model species’ niches or ecosystem roles) by providing many more data points.
> > - **Enabling new analyses**: With rich trait labels attached to images, researchers can study correlations between morphology and environment or behavior at scale.  For instance, you could analyze how wing shapes vary across climates, or link body color patterns to predation risk.  Traits explain ecological patterns better than just species names, and an automated pipeline makes these analyses feasible on large collections.
> > - **Boosting identification tools**: As shown with BioCLIP, trait-annotated images can improve automatic species-identification models.  Models trained on trait captions learn more nuanced visual cues, making them more robust to new specimens or image conditions.
> >
> > Overall, our pipeline provides a scalable way to inject expert-like knowledge (descriptions of body parts) into machine learning without manual annotation. By turning images into meaningful trait statements, it bridges the gap between digitized specimens and quantitative trait databases, supporting a wide range of biodiversity and ecological research. We have added this discussion to Appendix K and included a reference workflow in the anonymous code repository: https://anonymous.4open.science/r/morph_trait_annotation-8016.

---

### Official Review · Reviewer_dwm8 · 2025-10-31

**Soundness:** 3
**Presentation:** 2
**Contribution:** 3
**Rating:** 6
**Confidence:** 4

**Summary:**

This work uses sparse autoencoders to generate trait descriptions of insect images from BIOSCAN-5M. The authors select a 19k image subset of the dataset labeled to the species level and generate trait captions. They perform a series of ablations to explore how the train description vary as a function of workflow components. Three domain experts provided ratings to assess quality. The authors investigate improvements in a zero-shot setting when using their dataset for fine tuning BioCLIP.

**Strengths:**

- The combination of SAE and MLLM is a new approach to auto-generating trait captions for biological images.
- The authors provide a number of ablations, as reviewed by domain experts, to assess the quality of the resulting annotations as a function of different elements of their pipeline.
- Overall writing clarity is ok, with a few confusing sections as noted below.
- The annotation pipeline is the most significant part of the work thought it will need lots more testing in different domains, even of biodiversity imagery.

**Weaknesses:**

- The ablation study appears quite robust, but is missing an exploration of what $t_{activation}$ does.
- The explanation of the zero-shot experiments need some clarification.
- The authors could spend some time discussing how their labeling strategy might impact a hierarchical framing of the classification problem. It seems they significantly reduce the amount of data available by requiring images to be drawn from the same taxonomic level.

**Questions:**

- Few of the terms in the equations in section 3.1 are defined. What do they represent?
- Line 200: what proportion of BIOSCAN-5M is annotated to the species level?
- How is $t_{activation}$ set? How does varying the threshold impact trait selection?
- Line 318: What is the rubric used for human quality rating? The scores are stated without a range so it is difficult to contextualize improvements.
- In section 4.5, is 'in-the-wild' meant as natural images vs. the preserved samples in BIOSCAN-5m? Or something else? There are number of other distribution shifts inherent in the case of the former that worth expanding upon.
- How were the traits encoded in when fine-tuning BioCLIP?
- What is the species overlap with between BIOSCAN-5m and the BioCLIP training sets?

---

> ### Author Response · Authors · 2025-11-21
> **Rebuttal by authors (1/2)**
>
> We thank the reviewer for their thoughtful assessment and for highlighting both the novelty of our pipeline and the value of our ablation and expert-evaluation studies. Below, we provide detailed responses to the reviewer’s comments and questions.
>
> > **(W1, Q3) Effect of t_{activation}**
>
> In our pipeline, $t_{\text{activation}}$ is the scalar threshold used to decide whether a latent unit is “active” on an image when building the species– and genus–level frequency tables (Alg. 1). Concretely, for an image with sparse code $g(z) \in \mathbb{R}^n$, we include unit $j$ in the active set $Z_i$ iff its value in sparse code is greater than $t_{\text{activation}}$ post-activation. Larger values of $t_{\text{activation}}$ therefore yield fewer, more confident activations per image (higher precision, lower coverage), while smaller values increase coverage at the cost of noisier units. We set $t_{\text{activation}} = 0.9$ in our pipeline, chosen based on manual inspection of the resulting active latents, which yielded semantically coherent units, corresponding to biologically plausible traits (Section 4.4).
>
>
> > **(Q1) Terminology in Section 3.1**
>
> We apologize for the lack of clarity in terminology. We will clarify that $W_{e} \in \mathbb{R}^{n \times d}$ denotes the SAE encoder matrix that maps the dense backbone representation $z \in \mathbb{R}^{d}$ to the pre-activation latent vector $u \in \mathbb{R}^{n}$, and $W_{d} \in \mathbb{R}^{d \times n}$ denotes the decoder matrix that maps the sparse code back to the reconstructed representation $\tilde{z} \in \mathbb{R}^{d}$. The encoder and decoder also include bias terms:$ b_{e} \in \mathbb{R}^{n}$ and $b_{d} \in \mathbb{R}^{d}$, respectively.
>
>
> > **(Q2) Proportion of BIOSCAN-5M with species-level labels**
>
> Thanks for highlighting this. Approximately 9.2% of BIOSCAN-5M is annotated at species level. We make use of the entire BIOSCAN-5M for SAE training but only the species-labeled part is used for generating the trait dataset. We have updated Sec. 4.1 in the manuscript to clarify this.
>
> > **(Q4) Rubric and scale for human quality ratings**
>
> As mentioned in Sec 4.3 and Appendix E, we use a **five-point rubric** for trait quality, where 1 = “Completely Incorrect— Hallucinated or wrong body part”, 3 = “Partially Correct — Body part correct; 1–2 traits vague or incorrect.,” and 5 = “Completely Correct — Body part correctly identified; all traits visibly match (color, texture, shape, size); no hallucinations.” Each of the three domain experts independently rates each sampled trait on this 1–5 scale. To account for differing use of the scale, we then apply per-rater mean normalization, rescaling each annotator’s scores so that their personal mean matches the global mean before averaging across raters.
>
>
> > **(Q5, W2) Meaning of “in-the-wild” and distribution shift**
>
> By “in-the-wild,” we specifically **mean natural field photographs as opposed to the preserved, microscope-imaged specimens** in BIOSCAN-5M. BIOSCAN-5M images are lab images of individual, pinned specimens captured under controlled lighting and background, whereas the Insects dataset consists of volunteer field photos of live insects interacting with flowers and foliage, often partially occluded, in diverse poses, backgrounds, and viewing distances. This introduces multiple distribution shifts (background clutter, illumination, pose, occlusion, and scale) beyond the lab setting. We have clarified this definition and expanded the discussion of these shifts in Section D.2.
>
> > **(Q6, W2) How are traits encoded when fine-tuning BioCLIP?**
>
> We fine-tune BioCLIP in a standard image–text contrastive manner, where the **text input is a caption that concatenates the species name with the trait description**. Concretely, we use prompts of the form “A photo of 'species-name' with 'trait-description'.” We added this discussion in Section D.2.

---

> > ### Author Response · Authors · 2025-11-21
> > **Rebuttal by authors (2/2)**
> >
> > > **(Q7, W2) Species overlap between BIOSCAN-5M and the BioCLIP training set**
> >
> > BioCLIP is trained on 10.4M images, of which **1.1M images (10.6%)** come from BIOSCAN-1M, a subset of BIOSCAN-5M. In terms of labels, approximately **7.8K species (out of 454K taxonomic labels, ~2%)** in the BioCLIP training set correspond to BIOSCAN-1M.
> >
> > > **(W3) Labeling strategy**
> >
> > We thank the reviewer for raising this point. In the current work, we focus our trait-discovery and evaluation on the subset of BIOSCAN-5M with species-level labels, which indeed reduces the raw number of available images compared to using all taxonomic levels. This choice was deliberate: **our species-contrastive ranking relies on salient traits that are better expressed for a given species relative to its congeners, which requires instances at the same taxonomic level**.
> >
> > At the same time, our ranking procedure is already hierarchy-aware in that it explicitly contrasts species against their genus background, and the resulting traits can, in principle, be re-used at coarser levels (e.g., genus- or family-level classification) by aggregating species-level traits. The pipeline itself is not restricted to species labels: one could run the same procedure at any taxonomic rank (e.g., “species vs. genus”, “genus vs. family”) to obtain traits that are expressed at that level, or even construct a hierarchy of traits from coarse to fine. We did not explore these hierarchical variants in this paper due to space and scope, but we will clarify in the revised manuscript that (i) our pipeline uses species-labeled images, and (ii) extending our approach to explicitly model trait discovery across taxonomic levels is an interesting direction for future work.

---

### Author Response · Authors · 2025-11-28
**Author Response to Reviewer Discussion**

We thank all reviewers for their thoughtful and constructive feedback, and for recognizing the novelty of our trait annotation pipeline (dwm8), the quality of the writing (npDF, nY1L), and the practical relevance of large-scale trait annotation for biodiversity research (npDF). We appreciate the emphasis on the interpretability and modularity of our pipeline (npDF) and on the importance of morphological trait extraction as an underexplored research topic (nY1L). We provide detailed responses to each reviewer’s comments and questions in our rebuttal, and have revised the paper to incorporate additional experiments, analyses, and writing changes, which are highlighted in blue:
- Section 4.6: Cost-of-use Analysis (npDF)
- Section 6: Limitations, moved to main text (nY1L)
- Appendix B: Additional MLLM (GPT-5-mini) alongside Qwen-2.5-VL-7B and Qwen-2.5-VL-72B for comparison in Table B.3 (npDF)
- Appendix I: Additional feature detector baselines (nY1L)
- Appendix K: Downstream ecology applications (npDF)
- Writing changes in Introduction ($\S$ 1), Methodology ($\S$ 3), Experiments ($\S$ 4), and Experimental Setup ($\S$ D.2) (dwm8, nY1L)

We also sincerely thank the reviewer nY1L for the positive discussion and for confirming that W5, W7, Q1, and Q2 (degree of supervision, limitations of SAEs, and figures) have been fully addressed. We also appreciate their acknowledgement of the additional feature backbone ablation (W2), and we have revised the paper accordingly. For W4, we are grateful for the suggested additional baselines beyond GradCAM (PCA, ICA, etc.), and we will add the corresponding citations and explicitly note this as an important direction for future work. *As the reviewer notes*, evaluating these baselines fairly would require re-running the entire trait-generation and human-evaluation pipeline (including MLLM captioning and expert rating) for each representation, which is the dominant cost in our workflow in terms of both GPU time and annotator effort. *(The deleted comments correspond to a copy of our original rebuttal that was reposted and found to be redundant after clarification from ICLR organizers)*.

---

### Meta-Review · Area_Chair_FQ7F · 2025-12-14

**Summary:**

The paper proposes a pipeline to label insect images with short concise descriptions focusing on morphological traits.  The pipeline consists of  1) using sparse autoencoders (SAE) to identify highly activated image regions corresponding to traits, 2) obtaining set of distinctive traits for each species, and 3) using a vision-language model to get concise descriptions of the selected regions.   Using this approach on 19K insect images from BIOSCAN-5M, the authors construct Bioscan-Traits, a dataset of 80K trait annotations (e.g. succinct descriptions of morphological properties). The dataset is shown to be useful for fine-tuning BioCLIP for improved zero-shot species classification on a different dataset (the INSECTS dataset).

Reviewers noted that the combination of SAE and MLLM is an well-thought out and new approach to automatically generate trait-based captions for biological captions (dwm8) and the paper is mostly well-written.
The AC agrees that proposed pipeline is interesting and the paper to be clear, and believe most of the concerns indicated by the reviewers have been addressed during the author response period and the revised manuscript.

Some minor suggestions to authors:
- Consider having consistent presentation of MLLM outputs for Section H vs other parts of the paper.
- L1067: "Insects dataset Ullah et al. (2022)" => "Insects dataset (Ullah et al. 2022)"

**Reviewer Concerns:**

Reviewers expressed the following concerns:
1. Some details were not clear to some reviewers including some details about experiments and some equation terms (dwm8)
2. Potentially weak ablations (nY1L) with some parameter choices not ablated (dwm8)
3. Choice of trait discovery at the species level (dwm8)
4. Limited coverage VLMs/LMMs for trait-generation (npDF) and limited vision backbones (nY1L)
5. Limited analysis of cost/latency (npDF)
6. Whether fine-tuning on the data can help improve other stronger models such as BioCLIP-2 (npDF)
7. Questions about how the pipeline would be used by end-users (npDF)

The authors addressed most of the reviewer questions and revised the manuscript accordingly.  The revisions included the addition of experiments with more LMMs, backbones, and ablations and evaluating against BioCLIPO-2, cost analysis, as well as discussions and clarifications.  The reviewer concerns has been addressed to the AC's satisfactions.

**Reviewer Scores:**

Overall reviewers are positive on the work, with two marginal accepts (dwm8,nY1L) and one accept (npDF).

As the authors did a good job of responding to reviewer concerns and questions, and updates to the manuscript, the AC believe that reviewers would have kept their positive ratings (potentially even raising their scores).

---

### Decision · Program_Chairs · 2026-01-26

Accept (Poster)